# Basic Orbit Design and Maneuvers for Satellite Constellations Deployed Using Momentum Exchange Tethers

**Ben Campbell** [1],*[ID] **and Lawrence Dale Thomas** [2][ID]

1 Department of Mechanical and Aerospace Engineering, University of Alabama in Huntsville, Huntsville, AL 35899, USA

2 Department of Industrial & Systems Engineering and Engineering Management, University of Alabama in Huntsville, Huntsville, AL 35899, USA; ldt0001@uah.edu

* Correspondence: bac0038@uah.edu

**Abstract:** This article describes a new alternative approach to satellite constellation deployment by incorporating momentum exchange tethers (METs). Traditional methods of deploying satellite constellations have limitations, typically involving costly propulsion systems and extended dispersion times. METs offer a novel solution by efficiently transferring momentum between tethered objects, reducing the need for onboard propellants and streamlining the deployment process. This article discusses orbit design and maneuvers for different mission architectures using asymmetrical and symmetrical tether release techniques to deploy satellites into designated orbits. In addition, a short walkthrough of designing one possible constellation is given, showing how quickly a MET-deployed constellation can be established in low Earth orbit (LEO). This work contributes to ongoing research investigating the applicability of METs in satellite constellation deployments, which could potentially be a new opportunity for MET technology to start seeing routine usage in the space environment, and also enable new constellation architectures that have not yet been realized.

**Keywords:** satellite constellation; momentum exchange tether; satellite deployment; orbit design; mission design; small satellite

## 1. Introduction

The space industry is experiencing a surge in new satellite constellations, some of which are much larger than those originally conceived in the 1990s, creating a new category commonly referred to as "mega-constellations" [1]. Despite advancements in satellite technology through the decades, the process of deploying these constellations still relies on traditional methods, such as orbital transfer vehicles or significant propulsion systems onboard each satellite, which still pose the same challenges for constellation operators, largely revolving around dispersing their satellites in orbit [2,3]. Momentum exchange tethers (METs), a non-traditional propulsion method, could be a solution to alleviate some of the difficulties associated with satellite deployment and dispersion. MET-equipped deployers or orbital transfer vehicles (OTVs) could be made to rotate payloads on tethers and release them, providing rapid ΔV changes without necessarily requiring propellant-based systems. Early research insights have determined that they could simplify and expedite constellation deployment, reducing deployment times from years to weeks.

For this new concept blending the practices of satellite constellations and METs, this article will explain some of the basic concepts and considerations involved, which could be used to support potential future constellations. While not an exhaustive explanation of all factors to consider, it nonetheless provides a collection of building blocks to help start designing such a constellation. This article includes insights specifically on tether-deployed constellation orbit design and a conceptual deployment walkthrough. After introducing the existing technology, challenges, and significance of this research, five different questions for orbit design are discussed, the answers to which help narrow down options that can be

selected for different constellation architectures. One of these options is then selected for a conceptual constellation of 12 satellites, deployed using one of the simplest approaches described here.

## 2. Existing Technology and Challenges

### 2.1. Satellite Constellations

A satellite constellation is a group of coordinated satellites designed for specific tasks, like telecommunications or Earth observation. Recent interest in constellations for applications like these has led to plans by companies like SpaceX, OneWeb, and Amazon to develop and launch large constellations, with others in development [4]. Additionally, several other constellations are currently being considered and developed by government-backed research and defense organizations. These systems are typically placed in orbit either one by one on separate launch vehicles, or in batches on a single launch with additional propulsion for spreading them out later. For missions involving multiple satellites per launch, the usage of OTVs is an additional option to consider. Regardless, these options are costly in terms of finances, materials, and time, especially for smaller organizations and academia [2,4,5]. Each approach presents challenges for constellation operators with factors that can be complex to consider and settle on a solution for, and further become even more difficult to handle when dealing with logistics and execution. The ongoing research described in this article is primarily focused on constellations constructed from batches of smaller ride-sharing satellites as opposed to larger single satellites with dedicated launchers. Some challenges associated with ride-sharing satellite delivery for satellite operators include the following:

- Flying with an OTV spacecraft offers much more flexibility, but also relies on interaction with the OTV provider, and can again be costly due to the inherent specialization of the orbit delivery process of multiple onboard spacecraft. If the propulsion system included onboard the OTV is electric based or some other low-thrust system, the time to fully deploy each constellation spacecraft can be lengthy due to the low acceleration and resultant time to achieve $\Delta V$ to spread properly.

- Including onboard propulsion systems in a single-launch constellation offers more flexibility, but also comes with design and operation challenges. Some teams may not be able to handle the complexity, mass, and cost of these systems. Ensuring their reliability and on-orbit operation is another challenge on top of this.
  - While chemical propulsion is impulsive and can be used for rapid deployments, their relatively inefficient usage of mass can make them incompatible with some small satellite designs.
  - More efficient propulsion solutions like electric propulsion are available, and are more commonly used on constellations, but require more extensive dispersion times, typically in the order of months.
  - For even smaller satellites where chemical and electric propulsion systems are too large, differential drag techniques can be used for dispersion, but this method's performance is irregular, limits application to low altitudes with enough atmospheric interaction, and contributes to orbit decay, reducing satellite lifetime.

### 2.2. Momentum Exchange Tethers

Tether technology in space is not a very common practice, but it has been seen on a variety of missions. The first demonstration of this technology dates back to NASA's Gemini XI mission in 1967, which incorporated a 30 m tether connecting the crewed Gemini spacecraft with the Agena target vehicle. This system rotated and induced a light artificial gravitational acceleration for roughly 3 h, then disconnected and sent Gemini and Agena in separate opposing directions as part of a tether dynamics experiment [6]. Since this time, most missions incorporating tethers have been focused on tether dynamics similar to Gemini XI, electrodynamic tethers which rely on interactions with Earth's magnetic

field for propulsion or power generation purposes, or "space webs" where payloads are connected by tethers for formation flying purposes [7,8]. Momentum exchange tethers have seen some, but ultimately little usage in the space environment, and have also been used for individual deployments rather than multiple, as is being investigated by the research this article introduces.

METs are a space propulsion method that rely on the exchange of momentum between two tether-connected objects, typically a payload and a deployer or counterweight. They are not to be confused with electrodynamic tethers or space webs, which operate in different ways. The MET concept involves the mechanical rotation and separation of a spinning system, converting angular momentum into linear momentum upon release, which can alter the objects' velocities and positions [7,9,10]. Rotation in MET systems could be induced using either propulsion systems like cold gas thrusters (as used on Gemini XI), or electric motorized systems [11,12]. Operating this system is similar to a shepherd's sling spinning and releasing a stone, like in the Biblical story of David and Goliath. Generalized visualizations of such systems' operations are included later in this article. A key advantage of METs is their minimal fuel consumption (or complete avoidance thereof), making them attractive for missions with fuel supply and storage constraints. Space tethers can serve various purposes beyond propulsion, including station-keeping, attitude control, power generation, and passive deorbiting, although this article focuses solely on the tether-based deployment of constellations, particularly orbit and spacecraft design. Despite their potential benefits, METs require precise spacecraft-tether coordination and can be affected by environmental factors like solar wind, atmospheric drag, thermal effects, and micrometeoroid impacts over extended periods. Their payload capacity is constrained by mass and geometry, limiting their suitability for certain high-load missions or missions requiring higher spin rates or torques than can be achieved by feasible drive systems [13].

METs have been tested in a very limited capacity in past space missions, and traditional propulsion systems have remained dominant due to their broader applicability [7,9]. As a result, the technology readiness level (TRL) of MET technology can be estimated to be around level 6 or 7, where the technology has been demonstrated in the operational environment of space, but has not yet been applied outside of experimental research and development. METs that have flown specifically for momentum exchange purposes were designed as exceedingly long tethers, such as the SEDS-1 (1993) and SEDS-2 (1994) missions which utilized 20 km tethers [14], and the YES-2 (2007) mission which utilized a 30 km tether [15]. These missions involved the usage of tether systems to deploy only individual payloads. Many other efforts have conceptualized tether systems of similar or even larger scale for applications such as skyhooks or even space elevators, but most of these are not quite yet feasible with present resources.

This article introduces one potential application that could benefit from using tethers which has not been previously researched: satellite constellations deployed by tethers. Tether usage here occurs early in the mission lifetime, with a rotation and release of multiple payloads into desired destination orbits. This research work aims to keep the design of any applied tether systems within immediate feasibility by utilizing conventional materials and significantly shorter tether lengths on the order of meters instead of kilometers, and also put theory into practice by demonstrating this technology in the space environment.

Currently, the TRL of a MET-based constellation deployment system can be estimated to be near a level 3 or 4 at the time of this article's writing (late 2023/early 2024), but the overall research effort aims to raise this to the level 6–7 range. This TRL-raising work will culminate into a technology demonstration mission called ADRASTEA that will demonstrate symmetric and asymmetric tether release operations in the space environment following a launch partnered with NASA scheduled for August 2024. Post-flight results from this mission, when they are available, will be discussed in future publications.

## 3. Significance of Rapid Tether-Based Constellation Deployment

Tether-based deployment can offer two key advantages: quicker commissioning timelines and, as a result, financial benefits as well. Faster timelines result from shorter post-launch dispersion, and potentially simpler technology development depending on the constellation producer compared to traditional propulsion system development. Shortened timelines are produced with the utilization of pseudo-impulsive release maneuvers that deliver velocity change quicker than low-thrust electric propulsion or passive drag, like that used on multiple constellations such as those by Starlink/SpaceX and Planet Labs. This shortened time from launch to operation in space can allow for faster profit realization by getting technologies working in space sooner. Cost savings can potentially come from eliminating the need for more complex propulsion systems and onboard propellant for initial post-launch $\Delta V$ boosts (although smaller systems for later station-keeping may be recommended). Tether systems can also be stowed compactly in form factors comparable with the form factors of traditional propulsion systems, but also do not require expendable propellant, making them suitable for small spacecraft like CubeSats.

This potential for faster and more affordable constellation establishment could see major uses both around and beyond Earth. If current constellations are ever damaged at large or made unusable, such as by large space weather events or human-caused coordinated strikes, these assets are costly to bring back online, and can suffer periods of irregular or unavailable coverage while in recovery. Using a tether-based approach could minimize the internal cost and timelines of constellation recovery to rapidly respond to network damage. These satellites could also be used for rapid deployment of private networks, where coverage could be quickly needed over specific regions. Expectedly, constellation establishment by traditional means could take even longer if trying to work around another celestial body such as the Moon or Mars. If a tethered framework were to be applied to future constellations, it could possibly be a significant accelerator in developing the infrastructure needed in establishing manned settlements beyond Earth.

## 4. Materials and Methods for Designing Tether-Deployed Constellations

### 4.1. Conceptual Design

4.1.1. Conceptual Example "SwingSat"

Current research is exploring the potential of METs to enhance satellite constellation deployment. The scope of this work primarily revolves around three major task areas given below.

1. What are some best practices for performing satellite constellation deployments with tethers?
2. How should constellation spacecraft made for deployment by tethers be designed?
3. How does the performance of tethered constellation deployment compare against traditional deployment methods?

Of these, task areas 1 and 2 have received the most effort to date, and work is ongoing relating to task area 3. A summary of results from investigating task areas 1 and 2 make up the majority of the rest of this article, focusing on orbital maneuvers and spacecraft design. Future work will be published upon further progress in all three task areas.

For this article, a conceptual satellite constellation deployed by tethers will be discussed, and referred to by the name "SwingSat". This constellation is not a system currently being produced, but is solely for referring to a MET-based system when discussed in parallel with existing systems. It is primarily to help give a name and a visual representation to the reader for a system utilizing the concepts discussed throughout this article. An example of this system has been produced in the form factor of a 3U CubeSat, which is composed of two 1.5U halves, one being a "driver" hosting a motor drive system that rotates the "deployer" half, which hosts one or more sets of symmetrical tether deployment and release mechanisms for a set of small payloads. In operation, this system could spin up both halves in opposing directions (conserving angular momentum), extend the tethers (in this case,

tape measure booms), then upon proper alignment, speed, and timing, release the tethers and payloads. A representation of this system is shown in Figure 1. Note that this design is purely conceptual and as a visual aid to assist in explaining the concept.

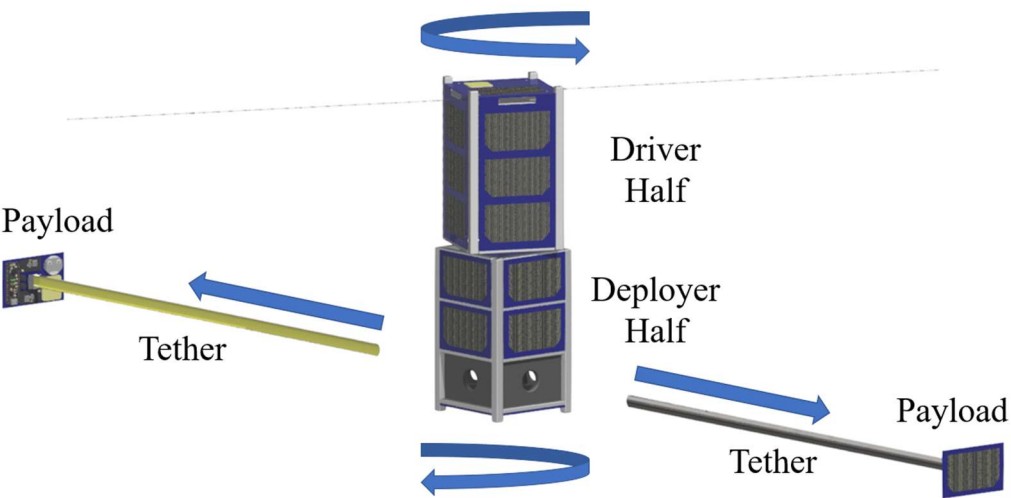

**Figure 1.** SwingSat concept as a 3U CubeSat.

### 4.1.2. Symmetric versus Asymmetric Release

The system uses paired tethers for stability upon release, as symmetrical release simplifies the return to a non-spinning state after deployment. Symmetrical release in this context involves releasing two opposing tethered objects simultaneously, which, if needed, can allow for payloads and potential counterweights of different masses and geometries as long as the center of mass remains near the central axis of rotation. The term symmetric is used to describe the geometry of the overall system, with a deployer in the center and two identical tethered payloads on either side of the deployer. Asymmetrical release involves letting go of just one tethered payload rather than two, which introduces complex disturbances due to a rapid change in the system's center of mass on one side, necessitating corrections with attitude control systems, which can be non-ideal if needing to restabilize between multiple sequenced deployments. An asymmetrical release can also alter the deployer's orbit after releasing a payload, which could require propulsive corrections depending on the nature of the mission. Symmetrical release, however, preserves the deployer's position, as the releases of opposing masses cancel out orbit-changing forces, assuming the released payloads' masses and tether lengths are identical.

With symmetrical releases, one aspect that must be considered is that releasing payloads into opposing directions means releasing payloads into different orbits. These two orbital planes of prograde and retrograde released payloads can be initially approximated as coplanar, but over significantly long periods, the effects of nodal precession can become visible due to the minute differences in altitude and velocities of the satellites in the two different planes. If these satellites include onboard propulsion for station-keeping; however, nodal precession effects can be easily mitigated.

### 4.1.3. Tether-Payload Release vs. Tether-Deployer Release

When considering tether release methods for the payloads, the tether can utilize a disconnection between either the tether and the payload, or between the tether and the deployer. Visual depictions of the pre-separation, separation, and post-separation states of such systems, for asymmetric and symmetric configurations, are given in Figures 2–13. It should be noted that these figures are generalizations of each release type, and the true exact locations of centers of mass are dependent on spacecraft design and may shift.

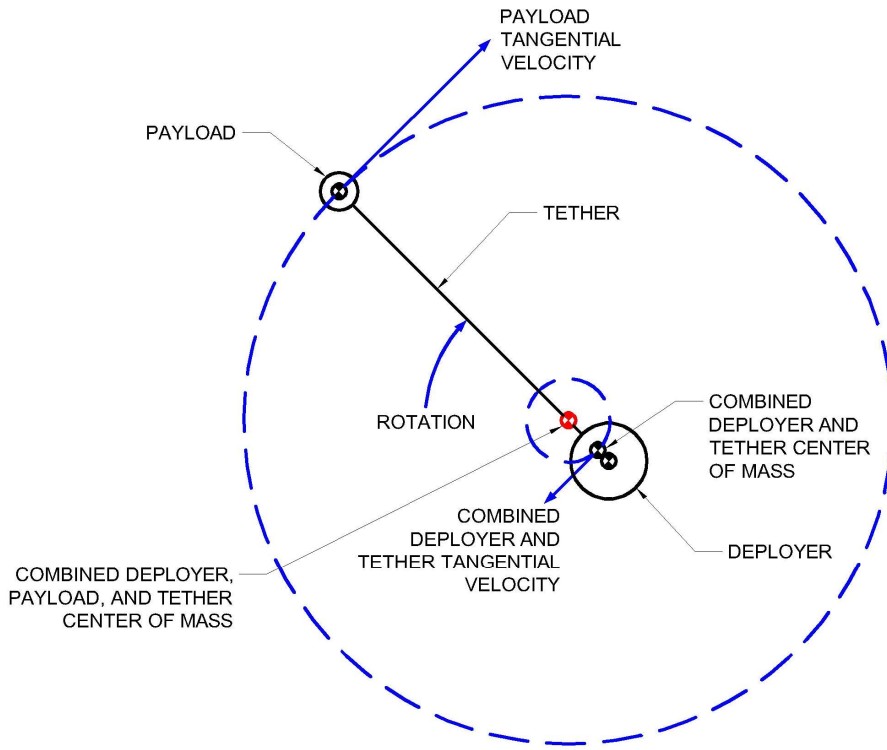

**Figure 2.** Asymmetric Tether-Payload Release Pre-Separation State.

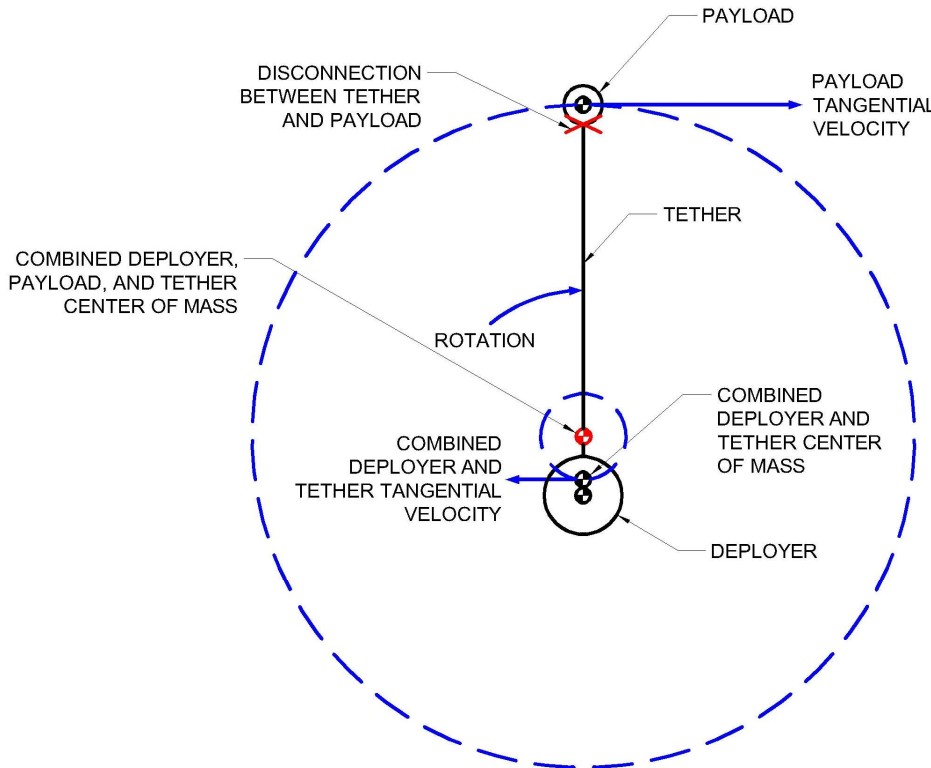

**Figure 3.** Asymmetric Tether-Payload Release Separation State.

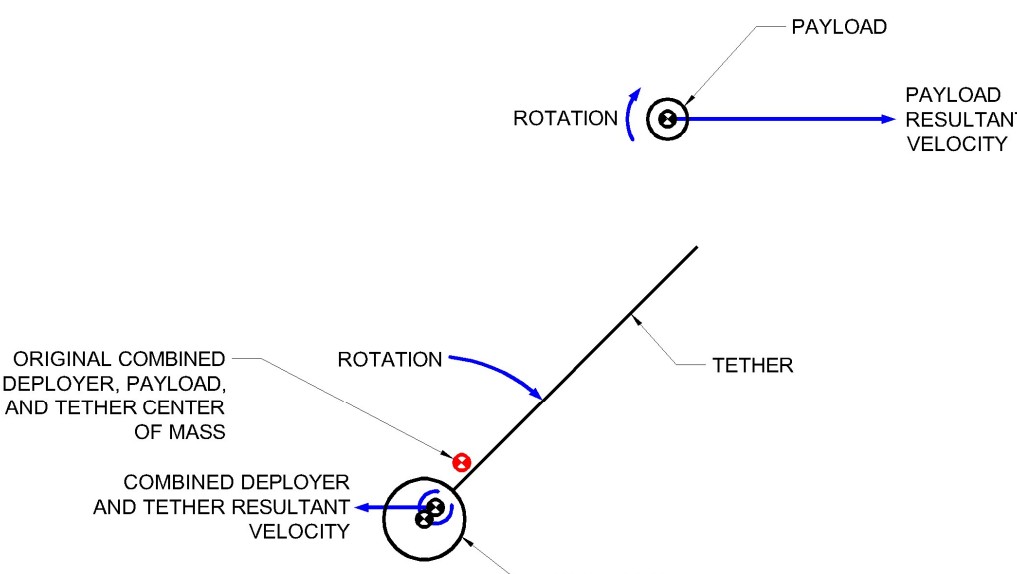

**Figure 4.** Asymmetric Tether-Payload Release Post-Separation State.

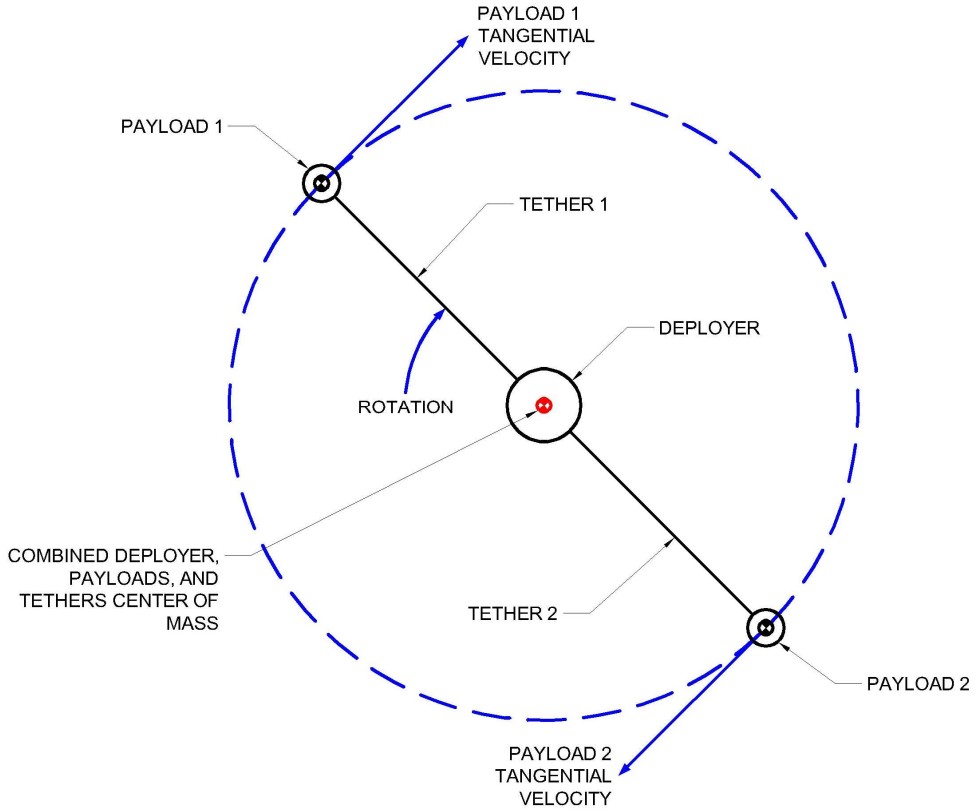

**Figure 5.** Symmetric Tether-Payload Release Pre-Separation State.

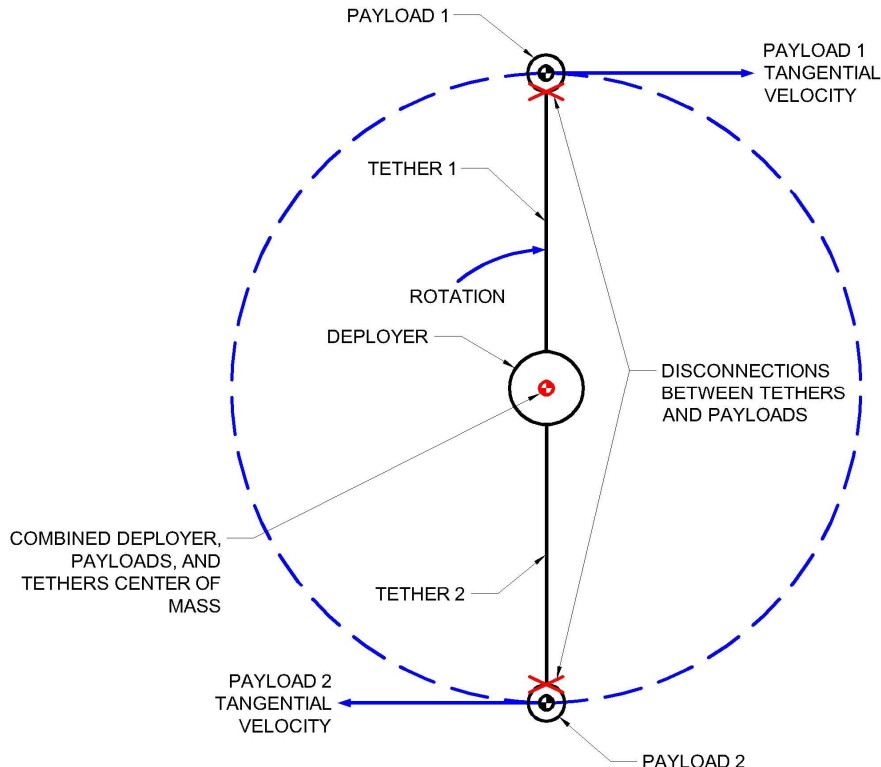

**Figure 6.** Symmetric Tether-Payload Release Separation State.

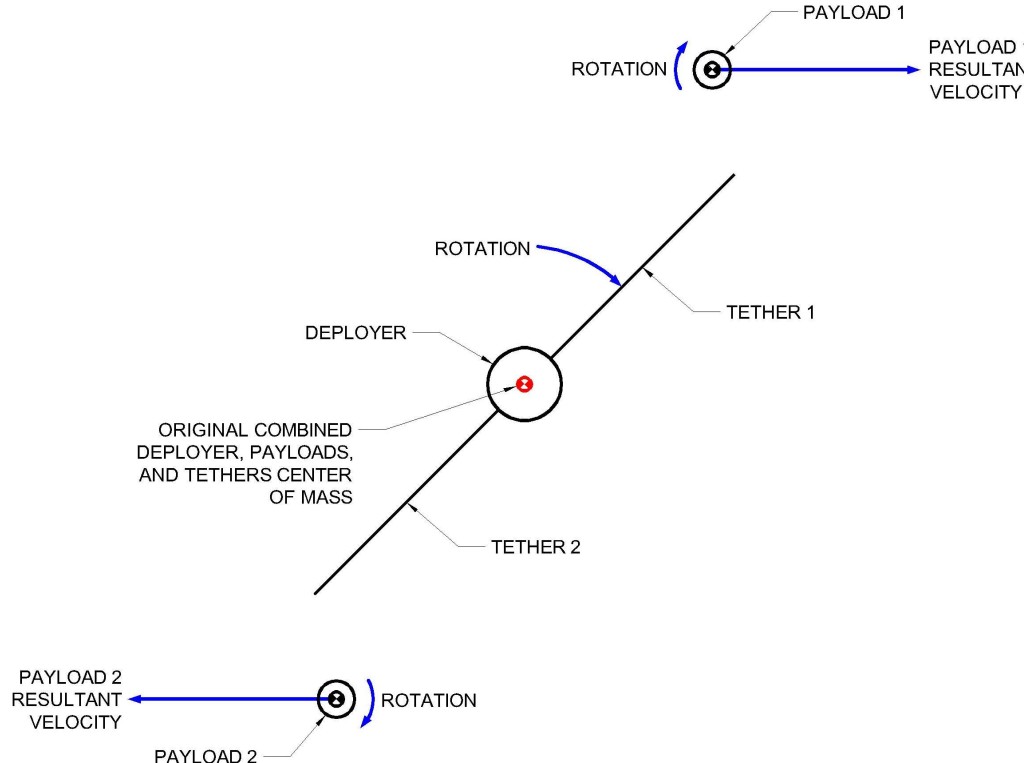

**Figure 7.** Symmetric Tether-Payload Release Post-Separation State.

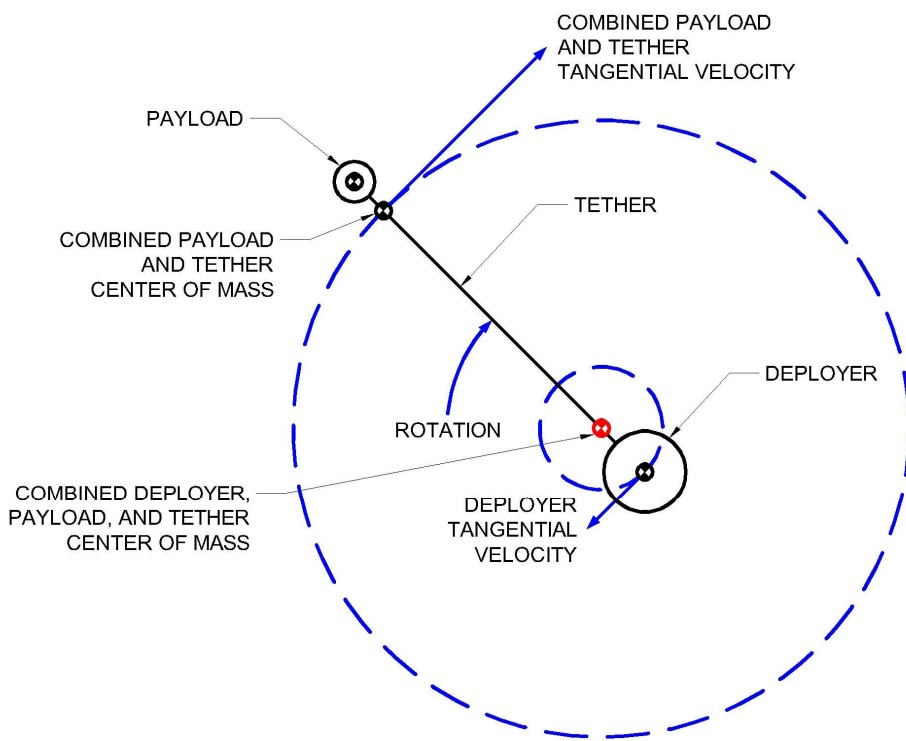

**Figure 8.** Asymmetric Tether-Deployer Release Pre-Separation State.

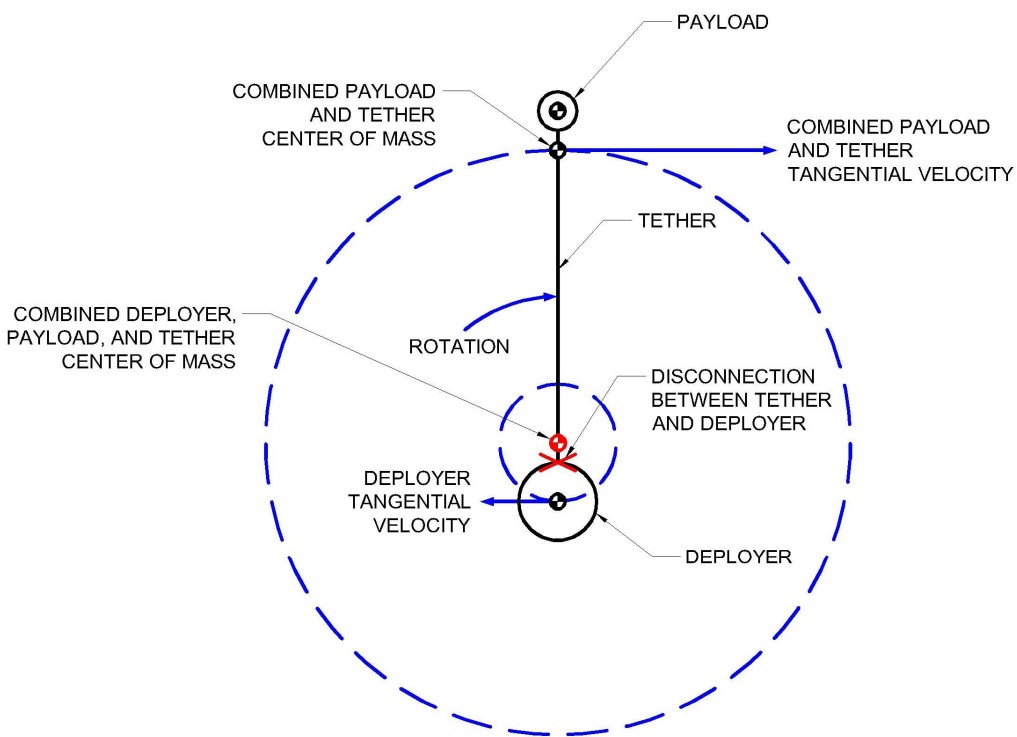

**Figure 9.** Asymmetric Tether-Deployer Release Separation State.

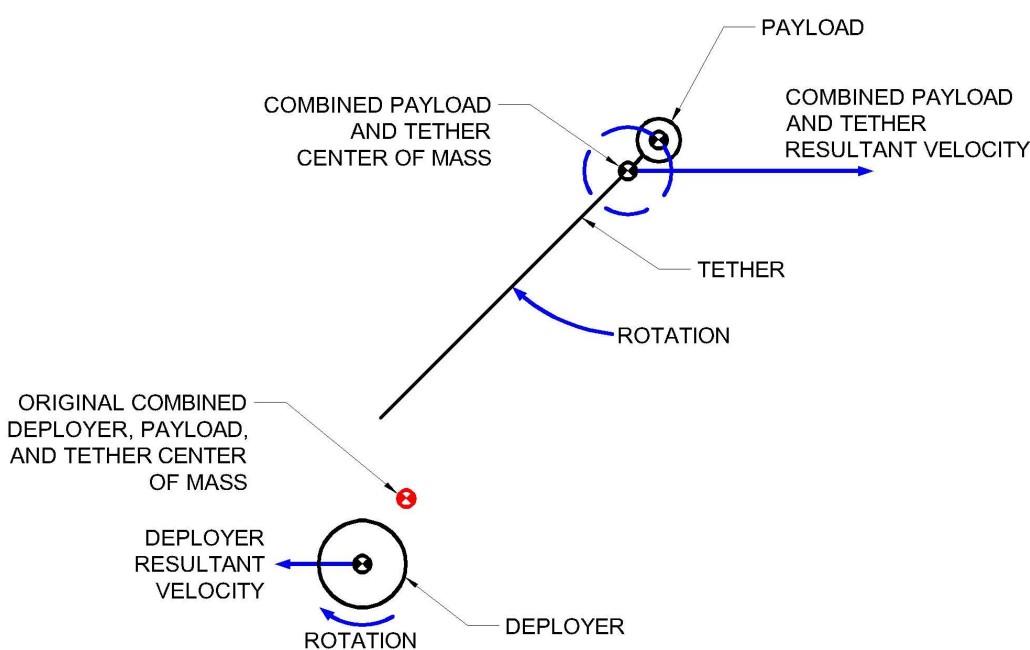

**Figure 10.** Asymmetric Tether-Deployer Release Post-Separation State.

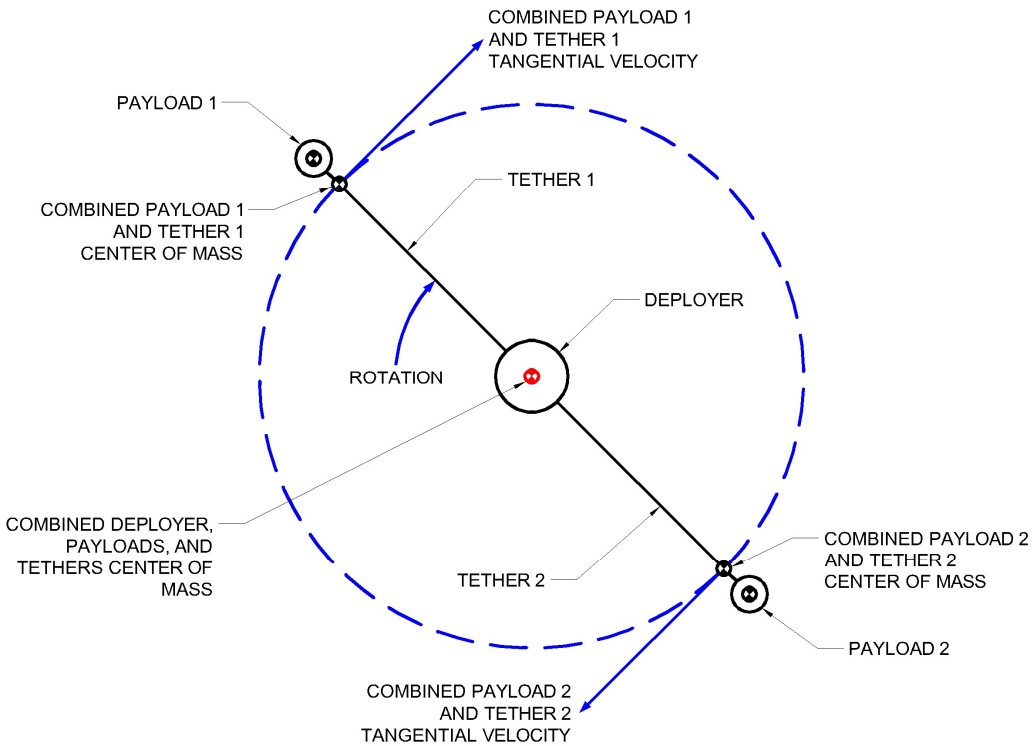

**Figure 11.** Symmetric Tether-Deployer Release Pre-Separation State.

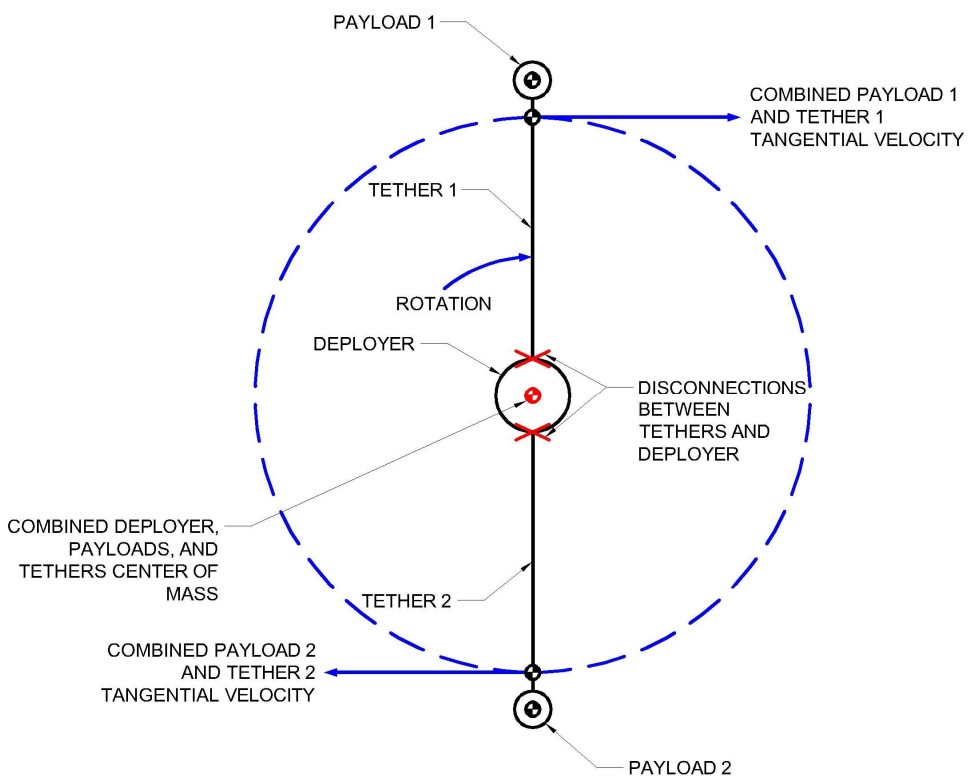

**Figure 12.** Symmetric Tether-Deployer Release Separation State.

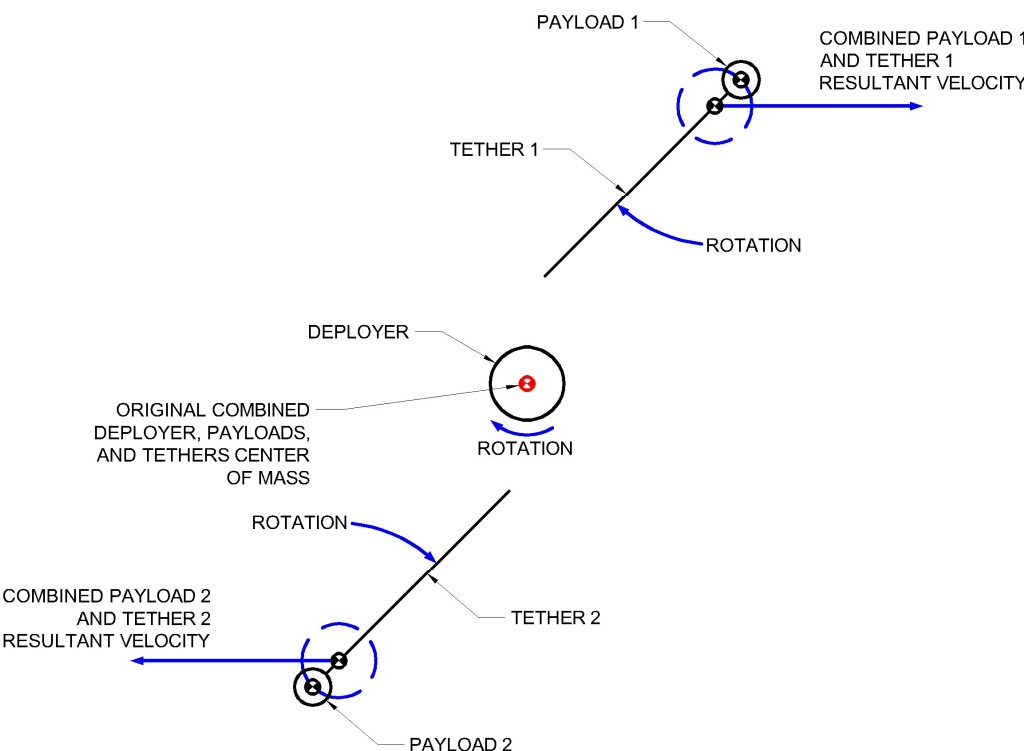

**Figure 13.** Symmetric Tether-Deployer Release Post-Separation State.

In designs separating the tether from the payload, the tether can be left with the deployer and leave no impact on the lifetime of the payload, aside from requiring a tether release mechanism onboard. Such systems are visualized in Figures 2–7. If this is pursued in a system utilizing symmetric releases, however, it is important that the timing of the prograde and retrograde payloads' release mechanisms are synchronized properly to avoid potentially unstable behavior that would result from delayed release sequencing. For example, if a prograde-released payload were one second before a retrograde payload releasing later off-nominally in the opposite direction, the one second period between releases would have the deployer and retrograde payload spinning in an unbalanced fashion like an asymmetric release system, and also result in the retrograde payload being released into an off-nominal orbital plane because of its angular displacement around the rotating system, thus also sending the deployer into a new off-nominal orbit as well.

To help combat the risks of non-synchronized symmetric tether-payload releases, it is possible to utilize tether-deployer releases instead. Such systems are visualized in Figures 8–13. With this method, the control of release between the deployer and tether can be controlled by a single system within the deployer, helping to ensure that both the prograde and retrograde payloads are released synchronously. An effect of this, however, is that the tethers would then remain attached to the payloads, which can be detrimental or beneficial to the payloads' operation and lifetime, depending on design and application. Negative effects of leaving the tether attached to a payload include an increase in cross-sectional area that increases atmospheric drag while in LEO (gradually decaying its orbit and reducing its lifetime), electrostatic charging while passing through regions of space with high magnetic field and radiation conditions, and additional mass and moment of inertia to handle while doing propulsive or attitude control maneuvers. While at first consideration each of these factors can be considered negative, some of them can also be reconsidered as opportunities for new functions of the payload. If the payload can actuate the tether's length, extending or retracting it on command, it can be used as a variable drag system for maneuvering and de-orbit operations. If the tether is conductive and able to be connected to other payload electrical systems, it can be repurposed into an electrodynamic tether as a power generation system and/or a low-thrust propulsion system.

*4.2. Orbit Design*

General constellation design entails choosing orbit parameters and satellite quantities specific to each application. For the sake of brevity, the basics of constellation design is not described in this article, but they are well-described in references [16–20]. In general, however, key factors to consider when designing a constellation include, but are not limited to, the following:

- Speed of communication between satellites and ground stations (affecting how fast information can be communicated, such as internet speed).
- Line-of-sight ranges (affecting how long ground links can last, such as length of a connection between one or more internet satellites before needing to connect to others to maintain connection).
- Revisit rate or frequency of repeated positioning of spacecraft over key locations (affecting how frequently location-specific data can be updated, such as monitoring the actively changing boundary of a wildfire).
- Range of coverage (such as limits in latitudes, affecting the limits of what regions, ground stations, and people can be covered).

Typically, constellations are constructed from one or more "rings" of satellites in orbital planes (such as a Walker constellation), but this is not necessarily always the case (such as with a Molniya-based constellation). These orbital rings are constructed in a manner where two or more spacecraft share an orbit and follow each other through the progression of their orbit path. These rings can be populated sparsely (such as with GPS), or densely (such as with Starlink).

When describing the orbits throughout this article, only simple elliptical orbits and in-plane maneuvers will be considered. While plane change maneuvers enabled by tethers are possible, they are not the primary focus of this article and will not be discussed in depth. The general geometry and parameters referenced from this point on are shown in Figure 14. Note that figures throughout this article depicting orbits have had their features exaggerated to assist in explanation, particularly regarding orbit radii and eccentricity. Some of the key orbit parameters to consider are orbit radius $R$, apoapsis $R_a$, periapsis $R_p$, semimajor axis $a$, true anomaly $\theta$, orbital period $T$, gravitational parameter of the central planetary body $\mu$, and the change in velocity needed to execute orbital maneuvers $\Delta V$.

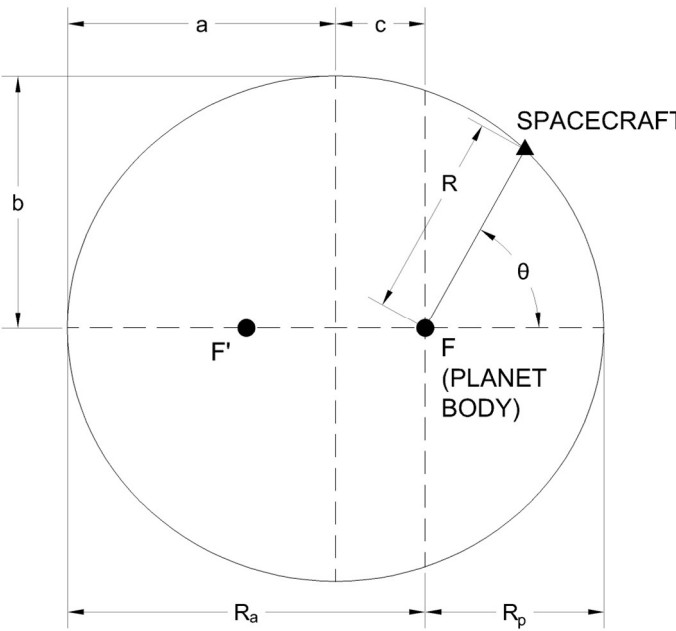

**Figure 14.** General orbit geometry and parameters, where a is the semimajor axis, b is the semiminor axis, c is the displacement of the planet/focus from the center of the ellipse, F and F' are the focal points of the ellipse (F being the planetary body), R is the orbital radius, Ra is the orbit's apoapsis, Rp is the orbit's periapsis, and $\theta$ is the true anomaly.

As the tether concept allows for near-instantaneous imparting of velocity change to deployed payloads, the largest orbit changes can be made by directing this $\Delta V$ in a direction tangent to the current trajectory of the payload-deployer system. It is also possible to direct velocity changes by tethers into directions other than directly tangent with the direction of orbital motion, but it is not as energy efficient when looking mainly to spread satellites immediately after launch. If, however, it desired to execute maneuvers such as plane changes, these maneuvers are still possible if the involved systems are designed and operated properly. These kinds of maneuvers are beyond the scope of this article, but will be addressed in future work.

This $\Delta V$ tangent to the direction of orbital motion can be directed prograde or retrograde to increase or decrease a payload's orbit apoapsis $R_a$ or periapsis $R_p$, thereby also affecting its semimajor axis $a$, eccentricity $e$, and orbital period $T$. A basic visualization of the orbits produced by prograde or retrograde release from a deployer are shown in Figure 15.

The process of deploying payloads into particular orbits forces some decisions to be made regarding the final orbit configurations of each payload and the deployer. For this system, there are currently five primary options to consider, each yielding their own benefits and drawbacks that can be selected for specific constellation design depending on application and the resources available to produce it. A flowchart describing these different options and the decisions leading to them is shown in Figure 16. Each orbit design question in the flowchart is described in the following five subsections. Future

research may discover additional constellation design options, which would be discussed in future publications.

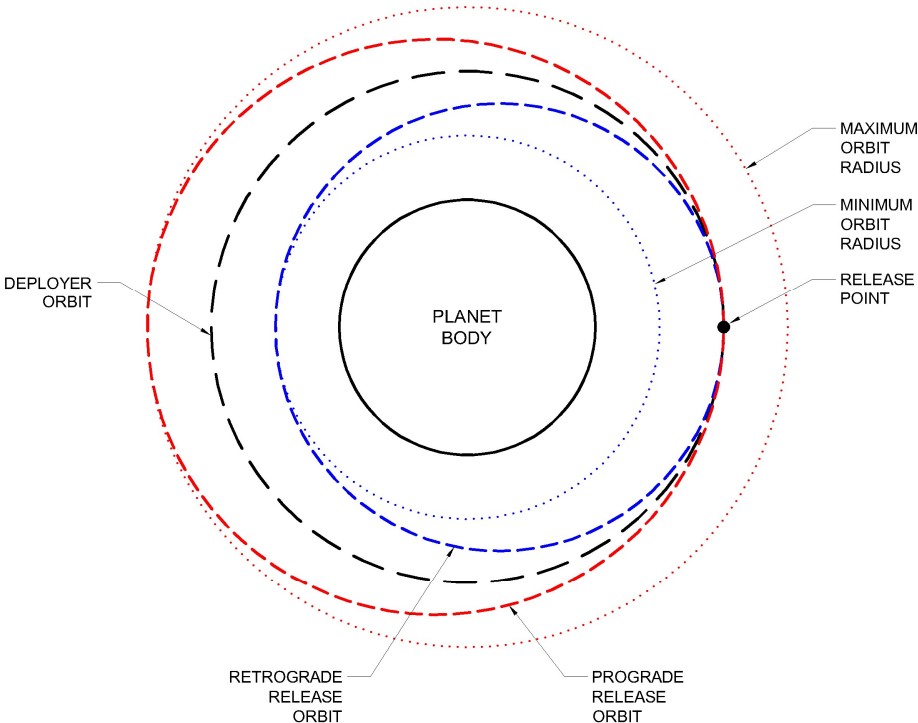

**Figure 15.** Basic SwingSat release orbits geometry.

### 4.2.1. Orbit Design Question 1: Single or Multiple Orbit Paths for Released Payloads

When considering single or multiple orbit paths, this asks whether payloads are required to be deployed solely into identical orbits with their only variation being in true anomaly. If only a single payload orbit path is permitted, this requires the system to utilize asymmetrical payload deployment, which would require either an onboard propulsion system to repeatedly boost the deployer's orbit back to its initial configuration to recover from payload momentum exchanges, or utilize symmetrical deployment with matched payloads and counterweights. The benefits of this asymmetric method entail very particular orbits with identical geometries, but the drawbacks come with devoting less resources to payloads and more to propulsion or counterweights, which reduce payload capacity and increase the number of launches needed to complete the constellation.

### 4.2.2. Orbit Design Question 2: Inclusion/Exclusion of Traditional Propulsion on Deployer for Asymmetric Release

When considering asymmetric release deployment, it is necessary to determine if the deployer spacecraft will include traditional propulsion, such as chemical or electric propulsion systems. If traditional propulsion is not included on the deployer, it will be necessary to include some other form of propulsion, such as deploying a tethered counterweight. Each of these options are feasible, but not advantageous for the overall intentions of tethered constellation deployments. Traditional propulsion utilizes established technology and does not contribute to space debris, but requires more design effort and the usage of propellant. Using tethered counterweights is technologically simpler, but is a poor use of spacecraft mass and volume and also contributes to space debris. If possible, these deployment methods should be avoided, but they are technically feasible options.

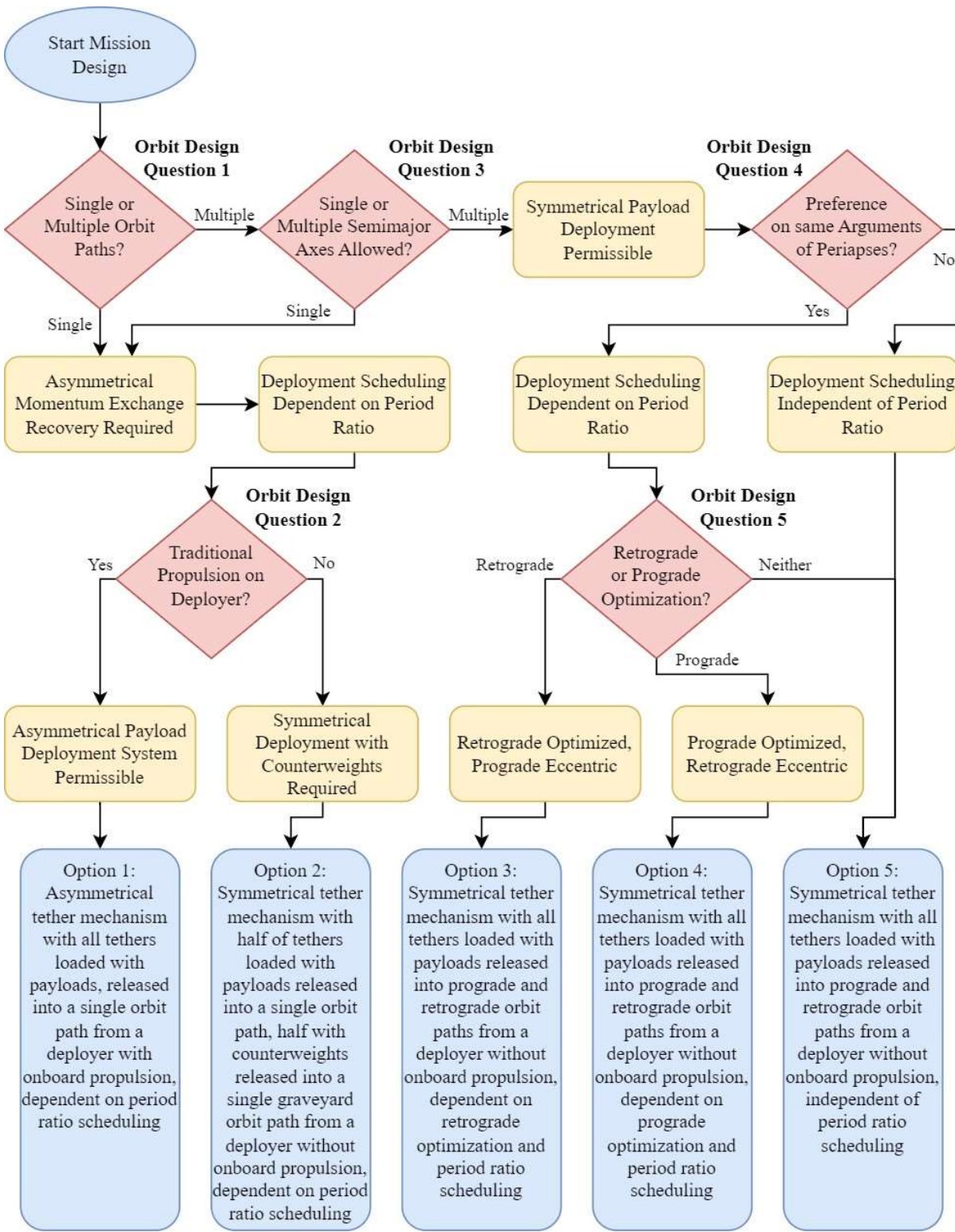

**Figure 16.** MET-deployed constellation design flowchart.

If the counterweights option were to be used with symmetrical releases, payloads would be released into either a prograde or retrograde release orbit, and counterweights released into the other. This particular approach is unfavorable, as it is not an efficient usage of overall mass and volume, which is critical for small satellites, and contributes to space debris. If this approach is required, it would be preferred to command the system to send payloads into prograde release orbits with the counterweights released into a

retrograde orbit small enough to put the weights on a re-entry trajectory for disposal to minimize space debris' contribution, potentially with a trajectory directly intersecting with the central planetary body. This approach with prograde-released payloads is shown in Figures 17 and 18, with options for circular or elliptical orbits, and similar graphics displaying retrograde-released payloads in Figures 19 and 20.

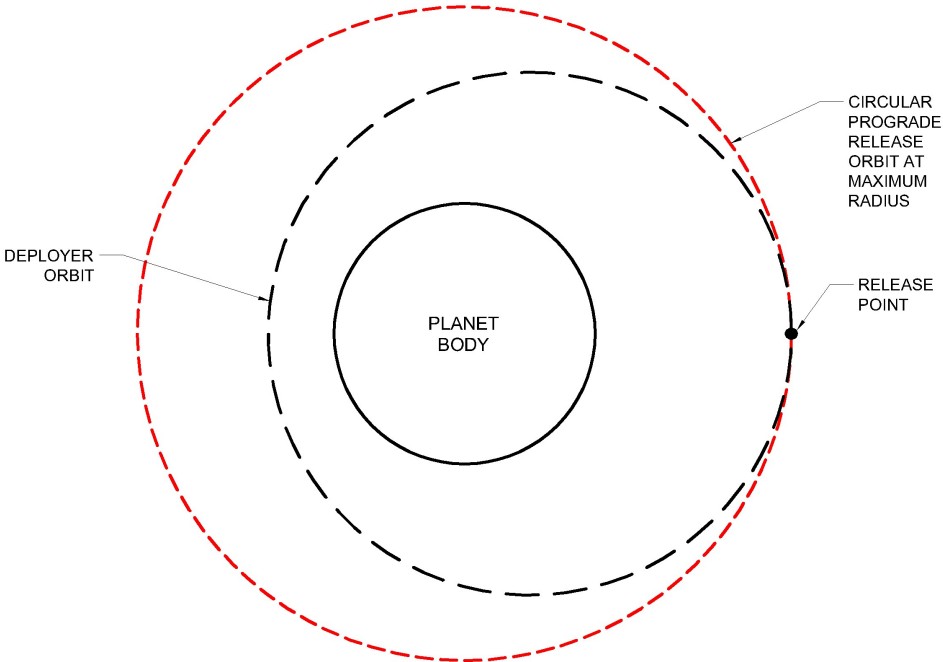

**Figure 17.** Prograde concentric circular release trajectory (retrograde orbit hidden).

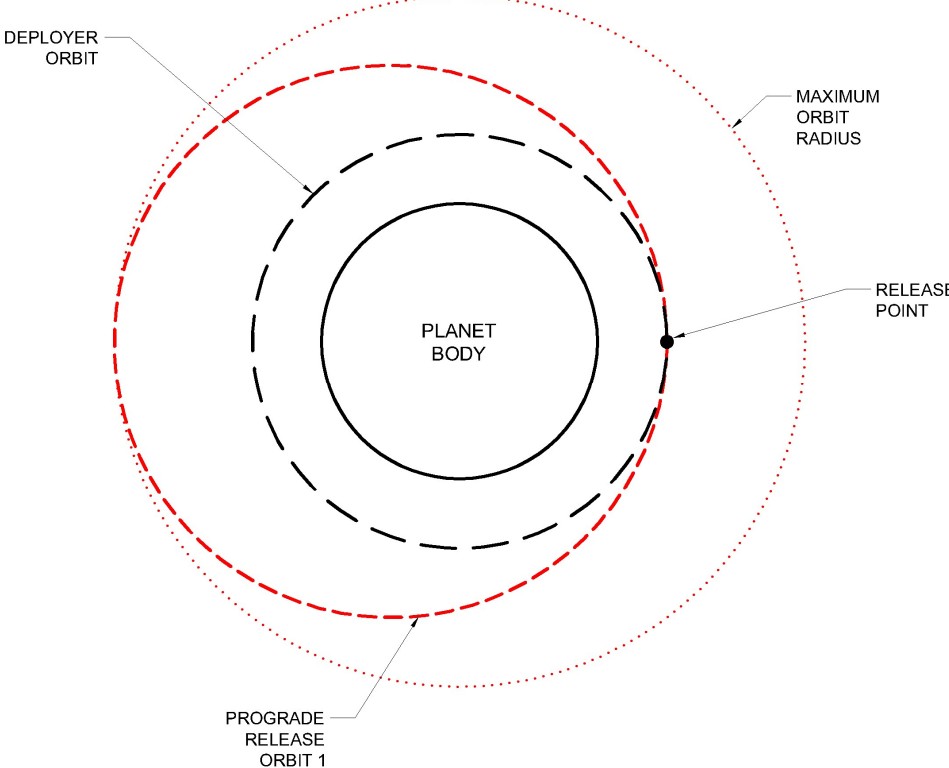

**Figure 18.** Prograde concentric elliptical release trajectory (retrograde orbit hidden).

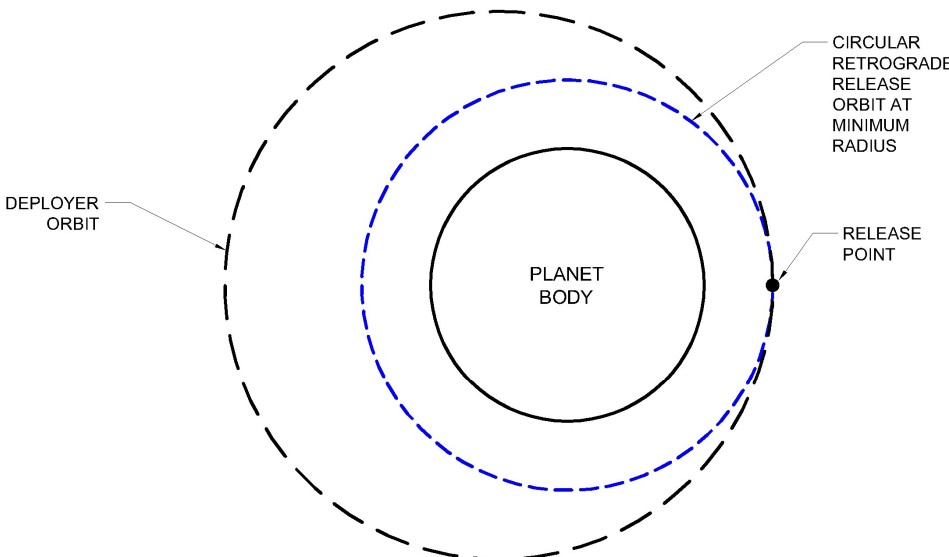

**Figure 19.** Retrograde concentric circular release trajectory (prograde orbit hidden).

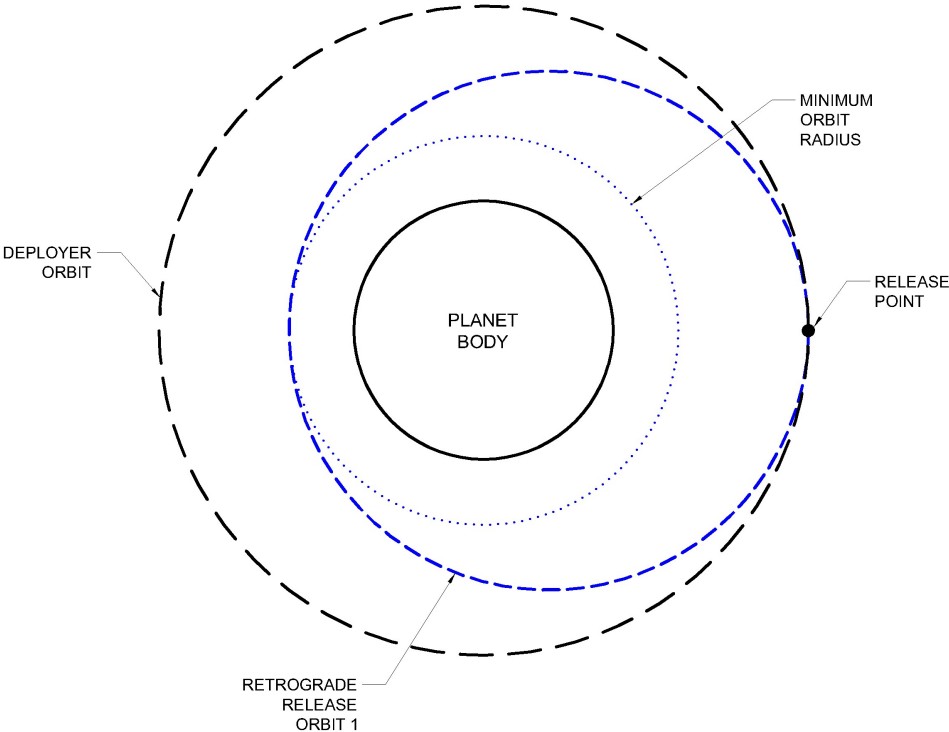

**Figure 20.** Retrograde concentric elliptical release trajectory (prograde orbit hidden).

### 4.2.3. Orbit Design Question 3: Single or Multiple Semimajor Axes for Released Payloads

When considering single or multiple semimajor axes being allowed, the key question is whether or not all payload orbits need to be dedicated to very similarly sized orbits. If this requirement is needed, this again results in requiring one of the asymmetrical payload release options. If there is allowance for variation in payload altitude however, symmetrical payload release becomes viable, and is a much more preferable option.

### 4.2.4. Orbit Design Question 4: Preference on Arguments of Periapses for a Prograde or Retrograde Payload Release Group

With the utilization of a symmetrical release, more options are available, which revolve around potentially optimizing the distribution of deployed payloads in either the prograde

or retrograde release orbits. This is not always necessary, but if it is, it would entail the deployer and either prograde or retrograde release orbits being configured in such a way that their orbital periods would allow for a uniform distribution of payloads along an optimized orbital ring, all with the same argument of periapses. The optimized payload group's orbits could be considered concentric, as they would be identical trajectories overlapping each other. It is possible that this optimized release payload orbit be made circular, but that would eliminate the need for a preference for the arguments of periapses of that payload group. Maintaining the same arguments of periapses would require the deployer to release payloads in the same direction at the same position in its orbit for each payload in the selected optimized group. Examples of these particular orbits are depicted in Figures 17–20.

The basic progression behavior of the released payloads naturally results in the prograde payload "lagging behind" the deployer's position after one deployer orbit due to the increased orbital period of the prograde payload. The reverse is also true for the retrograde payload "leading ahead" of the deployer after one orbit due to the shorter period for the retrograde payload. A visualization of this behavior is shown in Figure 21. As more payloads get released, this will naturally produce a chain of payloads that will appear to either "lag behind" or "lead ahead" of the deployer over time as the prograde and retrograde orbits become populated.

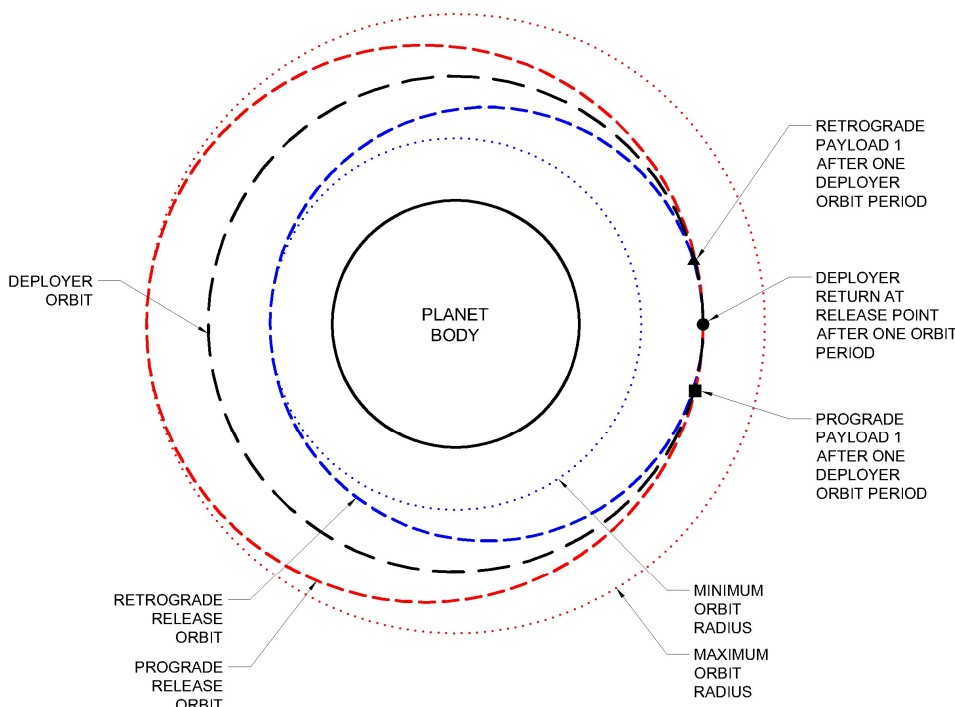

**Figure 21.** Basic payload orbit progression behavior post-release.

The primary benefit of the approach of optimizing one payload group is that it guarantees at least one payload group is released into a consistent pattern, uniformly distributed along a single release orbit path, but this comes at the expense of possibly producing gaps or overlaps in the distribution of payloads in the other release orbit and increasing its eccentricity, and can potentially take more time to execute when waiting for the deployer to intersect with the release orbits at the proper time to release each payload. A visualization of the different optimization options for prograde and retrograde orbit payload distributions and their resultant gaps or overlaps are shown in Figures 22 and 23. If this approach is used for optimizing specifically the prograde release orbit, the resultant geometry required for the deployer and retrograde release orbits should be carefully considered so as to not inadvertently release the retrograde payloads on a trajectory leading to a collision with the

central planetary body, unless the retrograde payloads are replaced with counterweights for an asymmetrical release approach as previously described.

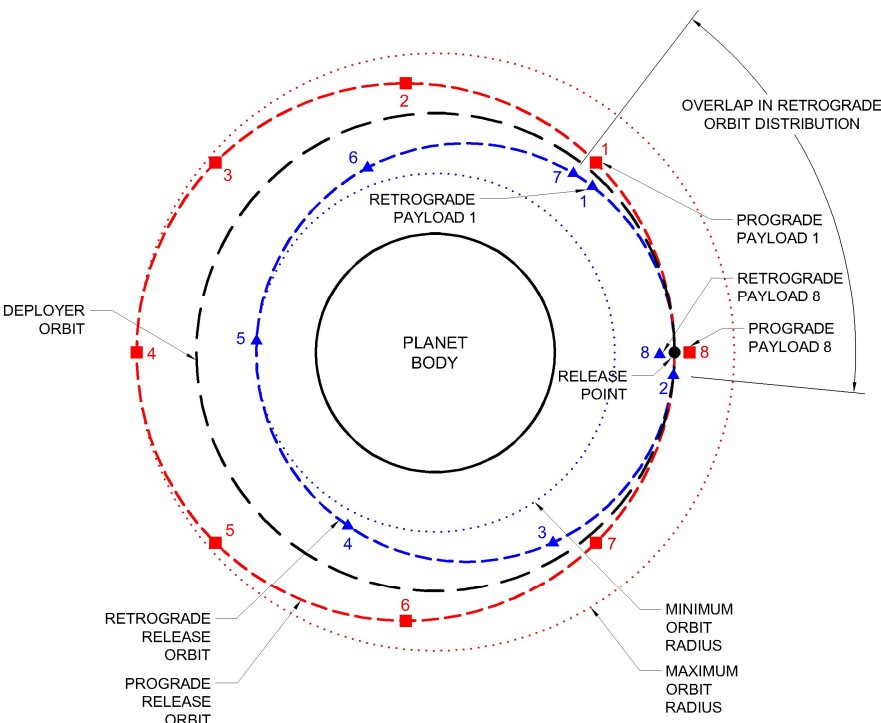

**Figure 22.** Basic payload orbit progression behavior with prograde release orbit distribution optimization for an example set of 8 prograde-released payloads and 8 retrograde-released payloads.

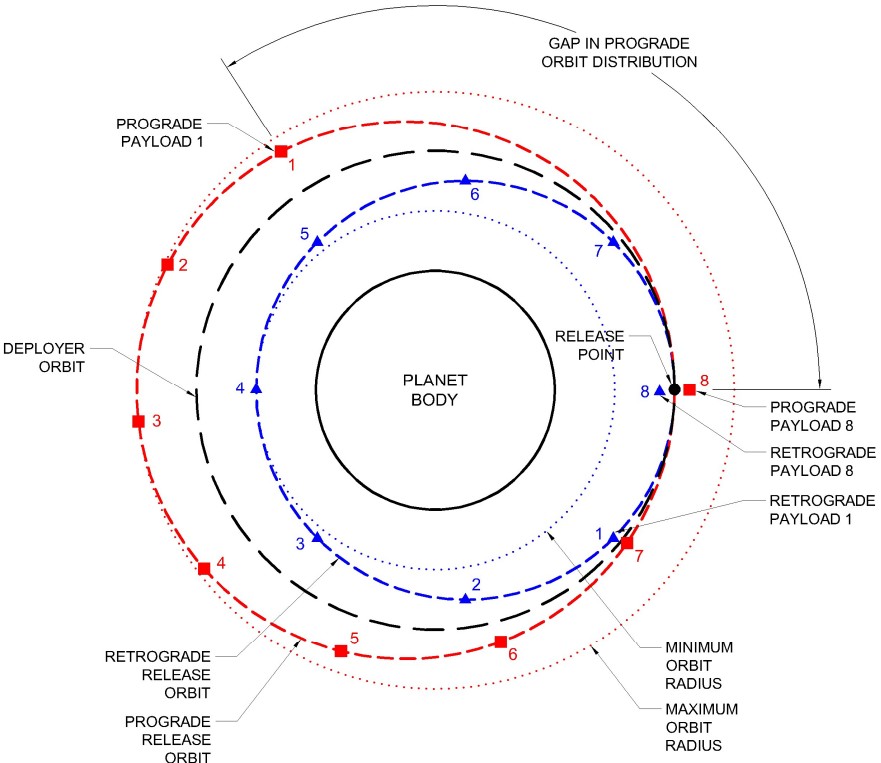

**Figure 23.** Basic payload orbit progression behavior with retrograde release orbit distribution optimization for an example set of 8 prograde-released payloads and 8 retrograde-released payloads.

### 4.2.5. Orbit Design Question 5: Prograde or Retrograde Release Orbit Optimization Dependent on Period Ratio

If choosing to optimize a prograde or retrograde release orbit payload group, orbit geometry and timing of the deployer's trajectory and release characteristics must be closely considered. This relationship between the deployer and optimized prograde/retrograde release orbit periods for proper release scheduling will here be referred to as the period ratio, $T_i/T_o$, where $T_i$ is the period of the innermost orbit for optimization, and $T_o$ the period of the outermost orbit for optimization. For prograde release orbit optimization, the innermost orbit would be the deployer orbit with the outermost orbit being the prograde release orbit. For retrograde release orbit optimization, the innermost orbit would be the retrograde release orbit with the outermost orbit being the deployer orbit. This can be visualized in Figure 24.

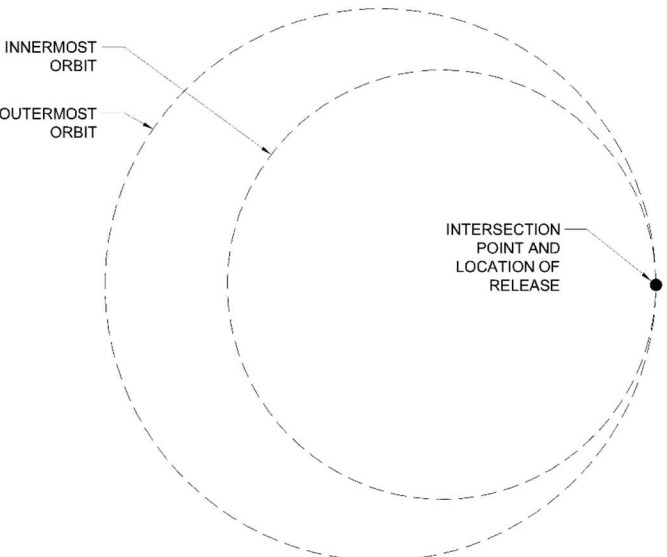

**Figure 24.** Basic inner and outer orbits.

The period of a single orbit $T$ can be defined by Equation (1), where and $a$ is the orbit's semimajor axis, $\mu$ is the gravitational parameter specific to the body being orbited, and $R_a$ and $R_p$ being the orbit's apoapsis and periapsis.

$$T = 2\pi\sqrt{\frac{a^3}{\mu}} = 2\pi\sqrt{\frac{(R_a + R_p)^3}{8\mu}} \tag{1}$$

The necessary period ratio $T_i/T_o$ for a selected pair of orbits containing a deployer orbit and an evenly distributed collection of payloads in a prograde or retrograde orbit is defined in Equation (2).

$$\frac{T_i}{T_o} = \frac{kn}{kn+1} = \left(\frac{a_i}{a_o}\right)^{\frac{3}{2}} = \left(\frac{(R_a + R_p)_i}{(R_a + R_p)_o}\right)^{\frac{3}{2}} \tag{2}$$

In this equation, $n$ is defined as an integer number of payloads to be uniformly distributed in the optimized orbit, and $k$ is defined as the integer number of instances where the deployer intersects the optimized prograde or retrograde orbit between releases. The lowest value for $k$ in this case is 1, which would result in the deployer releasing a payload every orbit. A value of $k$ equal to 2 would result in the deployer releasing a payload every two orbits, and so on.

With this in mind, it can be seen that there is a balance between speed of deployment and the geometry of the orbits (apoapsis, periapsis, and eccentricity) of that must be decided on when given a set number of payloads to be dispersed. As $T_i/T_o$ approaches 1, the deployer and optimized release orbits approach similarity, but this requires a high value for $k$ orbit intersections between releases, which would require a longer time to execute. Conversely, as $T_i/T_o$ approaches 0 (until $k$ is minimized to 1), the fastest time for dispersal is achieved with one release per orbit, but this then involves more dissimilar geometries between the deployer and optimized release orbits as a consequence of the largely different orbital periods, potentially resulting in high eccentricity for one or both of them. These factors must also be considered with the resulting $\Delta V$ requirements during deployment, where similar orbit geometries with slower dispersion times require much less $\Delta V$ at deployment than dissimilar orbits with more rapid dispersion times. More discussion on the determination of $\Delta V$ is included later in this article.

### 4.2.6. Neutral Release Orbit Optimization Independent of Arguments of Periapses

If the optimization of distribution of either release orbits is not required, these orbits can be produced in a manner allowing the payload orbits to have varying arguments of periapses, potentially leading to either circular or flower orbit configurations with one or more release points along the deployer's orbit. The existence of gaps when optimizing the retrograde release orbit and overlaps when optimizing the prograde release orbit implies that there are one or more intermediary "neutral" options that lead to complete coverage of both release orbits, but expectedly in a slightly non-uniform distribution pattern. The benefit of not optimizing one release orbit over the other is this more equal distribution quality between both orbits, and potentially quicker dispersion times thanks to not needing to wait for specific orbit syncing between the deployer and either prograde or retrograde orbits. These configurations are shown in Figures 25 and 26.

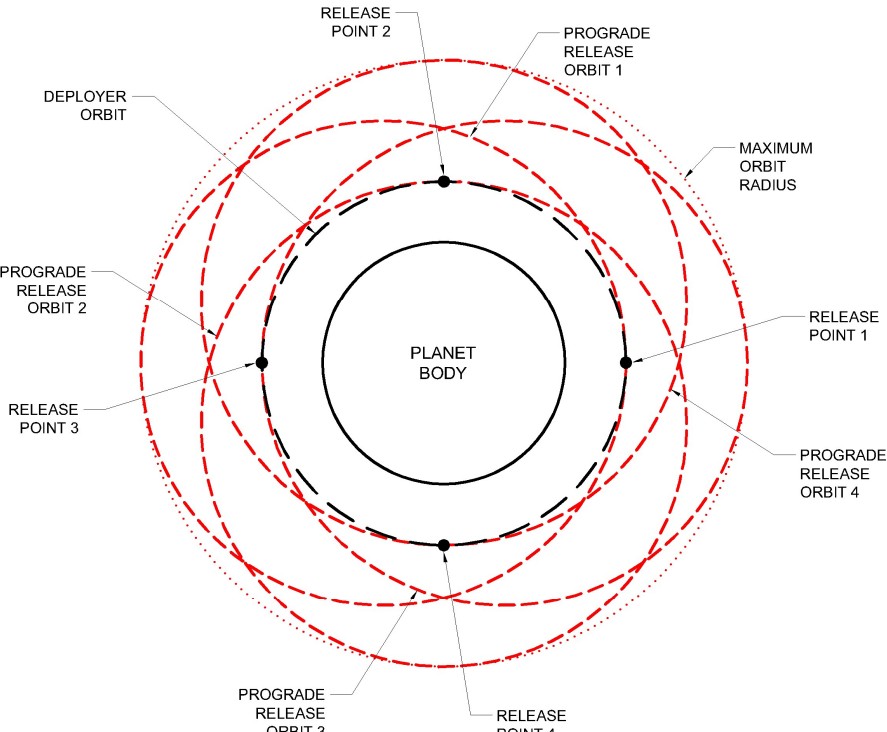

**Figure 25.** Prograde non-concentric elliptical release trajectories with 4 release points (retrograde orbits hidden).

There are several different approaches by which this neutral optimization can be pursued, but they are dependent on mission-specific orbit parameters and would need to

be determined on a case-by-case basis. Generally though, this can be done in a way that alternates priorities between prograde and retrograde releases to populate both release orbits. It is also likely advantageous to set the deployer orbit to be circular to allow for the most balanced distribution of minimum and maximum altitudes by each payload in their respective release orbits. A graphic depicting different regions for payload distribution with neutral optimization is given in Figure 27, and is primarily composed of geometries presented in their simplified forms in Figures 25 and 26.

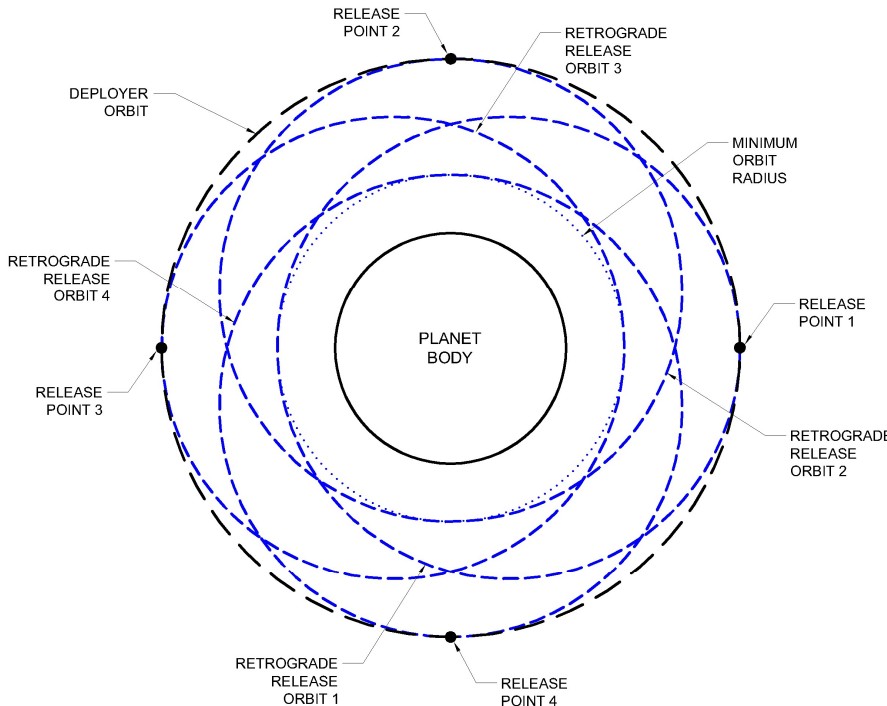

**Figure 26.** Retrograde non-concentric elliptical release trajectories with 4 release points (prograde orbits hidden).

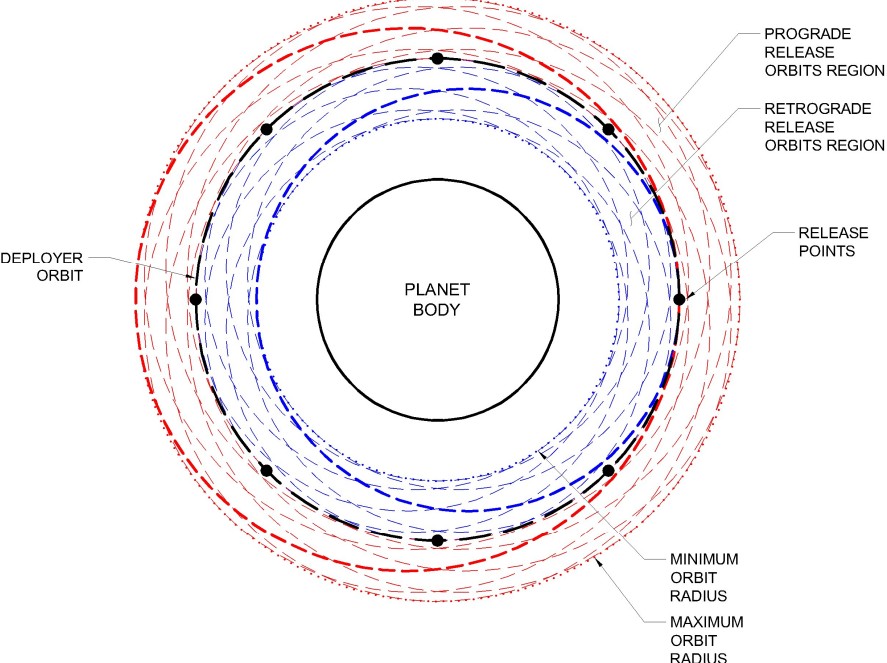

**Figure 27.** Basic payload orbit regions for neutral priority distribution optimization with 8 release points.

### 4.3. Conceptual Design Execution

Revisiting the SwingSat concept previously mentioned, the previously mentioned orbit design practices can be applied to a small theoretical constellation of 12 member satellites dispersed by a deployer spacecraft. This is just a small-scale example for the purposes of assisting in demonstrating an application of this work, and is not representative of a real constellation being developed. It is also focused solely on the generalized deployment process soon after launch, not later long-duration operations after initial dispersion. In this example, let there be 12 satellites that need to be distributed around a plane at a median altitude of 500 km. This constellation will be allowed to be deployed in a manner like design option #5 described in Figure 16. The tether deployment system will be designed to operate with the maximum $\Delta V$ imparted onto each deployed satellite being 20 m/s. The deployment speed is a fairly low and easily achievable using a tether deployment system for small satellite constellation deployment. Further details on deployer and tether design will be described in future published material by the authors. This future documentation is currently focused on tether deployment systems for satellites with individual masses of up to 500 kg, and necessary tape tether systems with masses on the order of a few kilograms made from conventional materials such as carbon fiber and spring steel.

Executing the maneuvers for satellite dispersion, the deployer will utilize neutral priority distribution optimization as depicted in Figure 27. This will be done by executing 6 symmetrical releases, each of which releasing two payloads at a time, adding up to the 12 total satellites to be deployed. For scheduling these releases, they will alternate priority for evenly spacing out satellites in the prograde and retrograde release orbits. For distribution of the satellites in each release orbit, the objective will be a nominal distribution of $360°/(12/2)$ satellites $= 60°$ between each satellite. The releases will be scheduled for an initial symmetrical release, then monitoring a difference in true anomaly $\Delta\theta$ between the deployer and first prograde-released payload until reaching $\Delta\theta = 60°$ to then execute the next release. This would be followed by monitoring $\Delta\theta$ between the deployer and first retrograde-released payload until reaching $\Delta\theta = 120°$ to then execute the next release. This process of alternating focus between the prograde and retrograde orbits for scheduling releases would continue for the full set of payloads to be deployed. A table describing the criteria for each release is shown in Table 1.

**Table 1.** Tethered payload release scheduling. Note that the first release is not preceded by any releases, hence not having an applicable true anomaly difference condition to trigger the release.

| Release Number | Bodies Considered for Scheduling | True Anomaly Difference $\Delta\theta$ | Date and Time of Release (UTC) |
|---|---|---|---|
| 1 | End of Deployer's first orbit after insertion | N/A | 1 October 2023 01:34:31.657 |
| 2 | Deployer and Prograde Payload 1 | $\left\|\Delta\theta_{Deployer} - \Delta\theta_{Prograde\ 1}\right\| \approx 60°$ | 2 October 2023 10:37:48.280 |
| 3 | Deployer and Retrograde Payload 1 | $\left\|\Delta\theta_{Deployer} - \Delta\theta_{Retrograde\ 1}\right\| \approx 120°$ | 3 October 2023 21:15:34.810 |
| 4 | Deployer and Prograde Payload 1 | $\left\|\Delta\theta_{Deployer} - \Delta\theta_{Prograde\ 1}\right\| \approx 180°$ | 5 October 2023 03:10:18.854 |
| 5 | Deployer and Retrograde Payload 1 | $\left\|\Delta\theta_{Deployer} - \Delta\theta_{Retrograde\ 1}\right\| \approx 240°$ | 6 October 2023 16:56:14.554 |
| 6 | Deployer and Prograde Payload 1 | $\left\|\Delta\theta_{Deployer} - \Delta\theta_{Prograde\ 1}\right\| \approx 300°$ | 7 October 2023 19:42:22.354 |

Following these criteria, these releases can be simulated in a software environment. In this simulation, the deployer was inserted into a simple 500 km altitude, 45° inclination circular orbit at 1 October 2023 00:00:00 UTC. The software that was used was STK Version 12.5.0 by AGI/Ansys, and idealized impulsive maneuvers were done using the Astrogator plugin. Orbit propagation was done using STK's Earth HPOP Default v10 propagator,

and factors of spacecraft mass, atmospheric drag, solar radiation pressure, and radiation pressure were neglected, as those factors are highly dependent on spacecraft design, and are assumed to have little effect on the results of this basic conceptual demonstration. More detailed analysis incorporating these kinds of effects on spacecraft with masses up to 500 kg each, along with varying tether designs, will be published in future work. The deployment system was assumed to operate in idealized conditions where the payload release vectors are directly tangent with the deployer's trajectory, and releases are treated as impulsive velocity change maneuvers independent of mass and size. While allowing the deployer and onboard payloads to propagate forward in time, and releasing payloads based on the release scheduling criteria previously described, the dates and times of potential releases could be determined, and are also included in Table 1. The release maneuvers deployed payloads away at exactly 20 m/s directly prograde or retrograde. Visuals of the state of the constellation at each of the release times are given in Appendix A.

The resulting constellation produces two very similar but separate orbital planes, each with six satellites roughly evenly distributed around them, and is shown in Figure 28. Payloads that were released into prograde directions are shown in red, retrograde payloads in blue, and the deployer in white. The prograde payloads were placed into orbits with an apoapsis altitude of ~559 km, and a periapsis altitude of 500 km, and the retrograde payloads were placed into orbits with an apoapsis altitude of 500 km, and a periapsis altitude of ~416 km. The altitude bounds of 416–559 km can be narrowed closer to the 500 km median altitude with a lower release velocity than the 20 m/s used in this analysis, but this comes with the consequence of longer time periods between releases, increasing the overall dispersion time. Conversely, more rapid dispersion periods can be achieved with higher release velocities beyond 20 m/s, but this comes with the consequence of a wider altitude range, where retrograde-released payloads can be sent to precariously low altitudes and experience increased effects of atmospheric drag.

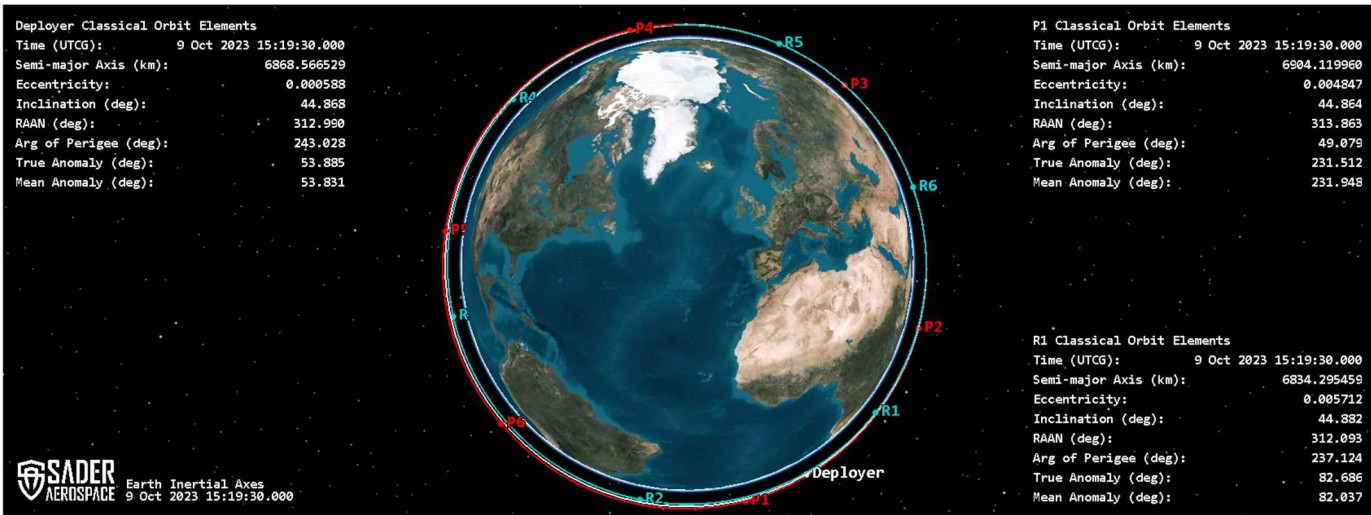

**Figure 28.** A 12-member constellation distributed in approximately 9 days post-launch (9 October 2023 15:19:30.000), simulated in STK software. Objects P1–P6 (red) indicate the first through sixth prograde-released payloads, and objects R1–R6 (blue) indicate the first through sixth retrograde-released payloads.

With each plane having different orbital periods because of their differing semimajor axes, there are occasional brief instances where satellites in two different planes can be in close proximity to each other in terms of true anomaly $\theta$, but this is soon followed by their separation as one passes by another. Despite this occasional "overlapping" behavior, due to the nature of the distribution, there will never be more than approximately 60° of separation between any two adjacent satellites in the combined planes. While not (yet) optimized for precise positioning of these satellites, this deployment process can effectively

distribute its satellites throughout the full spread of an orbital plane in the early phases of this particular conceptual constellation.

## 5. Discussion

This summary of different orbit configurations and release maneuver options is not exhaustive, but presents some key concepts to consider if a MET-deployed constellation were to be developed. Answering the five discussed orbit design questions can help in determining the nature of maneuvers needed to establish a constellation. These questions essentially all revolve around the similar theme of how similar or dissimilar the orbits for each member satellite are allowed to be, where missions allowing more dissimilar orbits can be deployed quicker and/or easier, while missions requiring more similar orbits can take longer and/or be more challenging to achieve. The deciding factors on whether these orbits can be similar or dissimilar are sourced from major requirements of the overall mission architecture itself, which is specific to the teams developing them and the proposed constellation's intended application(s). The selection of orbit design options can also inform the generation of requirements needed for associated spacecraft design.

The methods of deploying payloads by tether described here are not exhaustive, but give some initial insight to the thought processes that would likely be followed for designing such a system. The example given in this article is just one method (and the most chaotic or unoptimized of the five mission design options in Figure 16) but is still one viable means of dispersing satellites around an orbital plane for a constellation. A more optimized case would require some additional assistance in terms of attitude control and propulsion capabilities on the deployer, which would be easily achievable for moderately sized deployer spacecraft (such as ESPA class equivalent and larger).

## 6. Conclusions

In conclusion, this work has shed light on various orbit configuration and release maneuver options for the deployment of satellite constellations using METs. For newer satellite teams wanting to join in the constellation ecosystem, more feasible methods of establishing constellations can be helpful in this effort, with METs being one potential solution to consider. By implementing METs into constellation deployments, satellite teams can more easily handle design, resource, time, orbit delivery, and cost constraints that have been difficult in the past. The material discussed in this article gives a brief insight and demonstration of how METs and satellite constellations can be combined for rapid deployments to produce new mission architectures that could be flown. Future research and publications will include applications of the concepts described in this article to existing real-world constellations and comparing them against current industry norms, and also development and results from a small technology demonstration mission called ADRASTEA to be flown in the space environment. Current and future work in these areas will continue with the intent of improving the way constellations of any scale are created at key locations in accessible regions of space.

**Author Contributions:** Conceptualization, B.C.; methodology, B.C.; software, B.C.; validation, B.C.; formal analysis, B.C.; investigation, B.C.; resources, B.C. and L.D.T.; data curation, B.C.; writing—original draft preparation, B.C.; writing—review and editing, B.C. and L.D.T.; visualization, B.C.; supervision, L.D.T.; project administration, B.C. and L.D.T.; funding acquisition, B.C. and L.D.T. All authors have read and agreed to the published version of the manuscript.

**Funding:** This research is jointly funded by the NASA Alabama Space Grant Consortium (NASA Grant #80NSSC20M0044) and the UAH College of Engineering.

**Data Availability Statement:** No new data was created or analyzed in this study. Data sharing is not applicable to this article. Future supporting information (such as further publications and demonstration flight data) will be available at space.uah.edu and saderaerospace.com.

**Acknowledgments:** The author thanks his research advisor L. Dale Thomas for supporting this research, along with generous support from the NASA Alabama Space Grant Consortium and the UAH College of Engineering. The author also thanks the student team members from the UAH Terminus Spaceflight Research Group involved in the ADRASTEA mission for assisting in developing new technologies relating to the concepts described in this article.

**Conflicts of Interest:** The authors declare no conflicts of interest. The funders had no role in the design of the study; in the collection, analyses, or interpretation of data; in the writing of the manuscript; or in the decision to publish the results.

## Appendix A

This appendix includes visual representations at the times of different release maneuvers while dispersing payloads around an orbital region for a conceptual 12-member constellation. The states are sequentially included below.

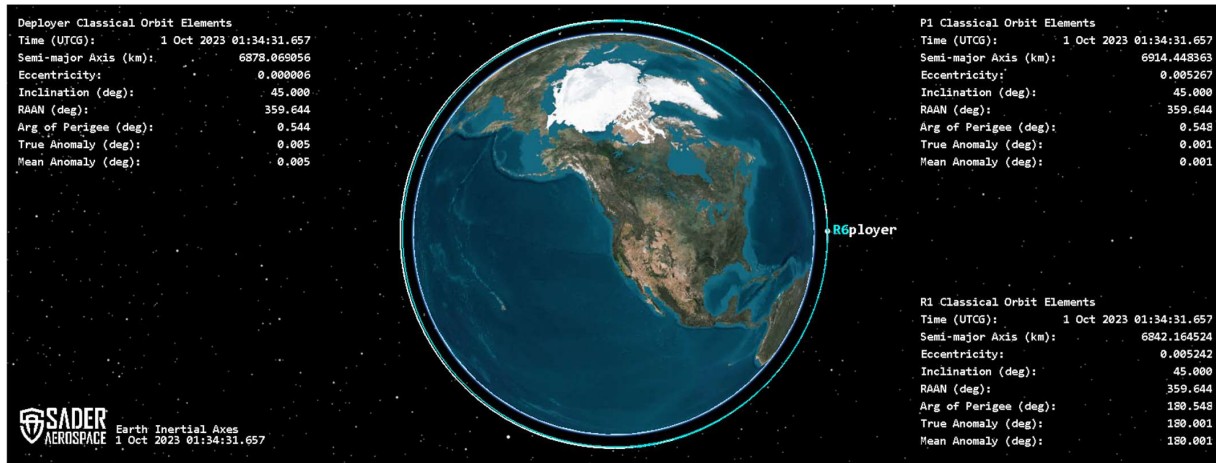

**Figure A1.** Constellation at the time of release maneuver 1 (1 October 2023 01:34:31.657).

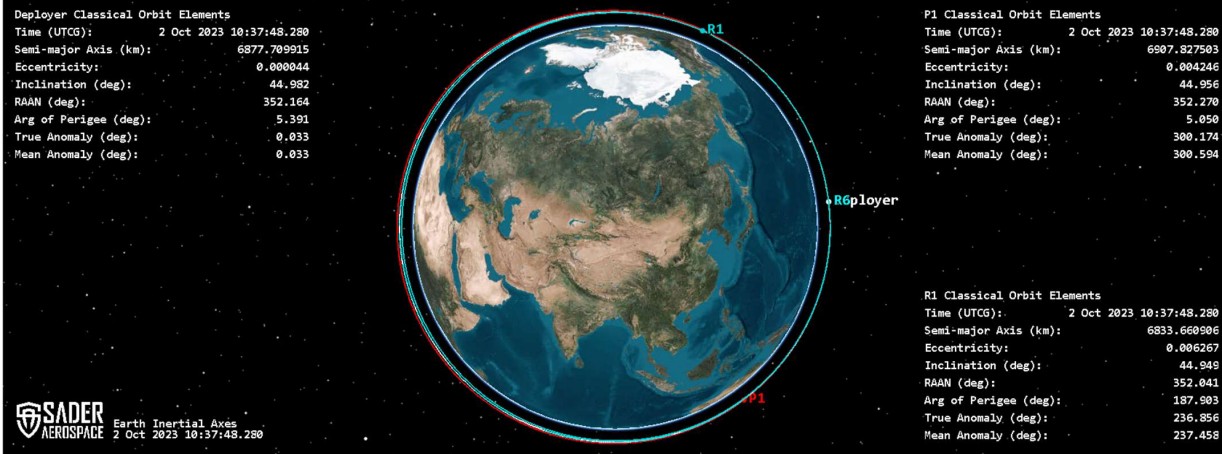

**Figure A2.** Constellation at the time of release maneuver 2 (2 October 2023 10:37:48.280). Object P1 (red) indicates the first prograde-released payload, and object R1 (blue) indicates the first retrograde-released payload.

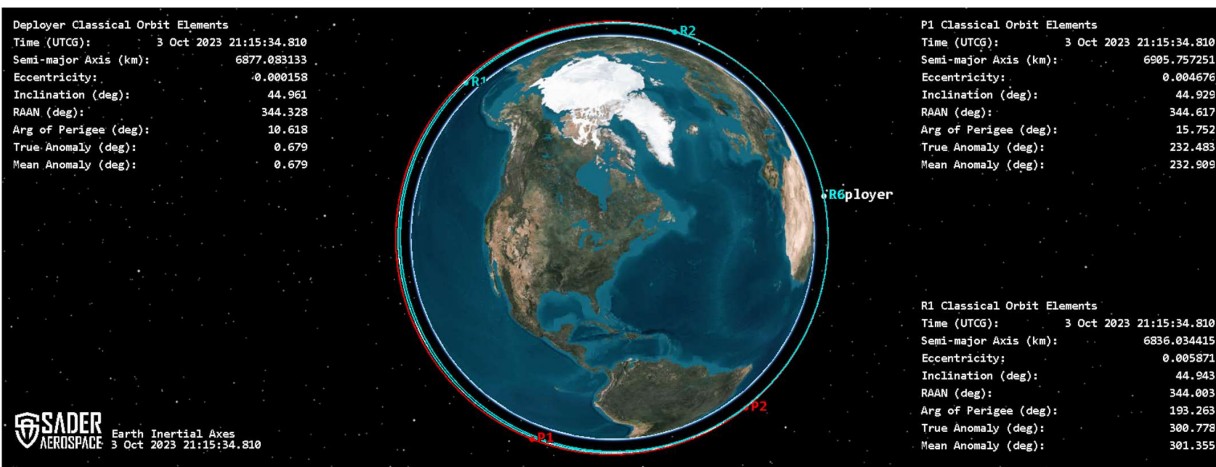

**Figure A3.** Constellation at the time of release maneuver 3 (3 October 2023 21:15:34.810). Objects P1 and P2 (red) indicate the first and second prograde-released payloads, and objects R1 and R2 (blue) indicate the first and second retrograde-released payloads.

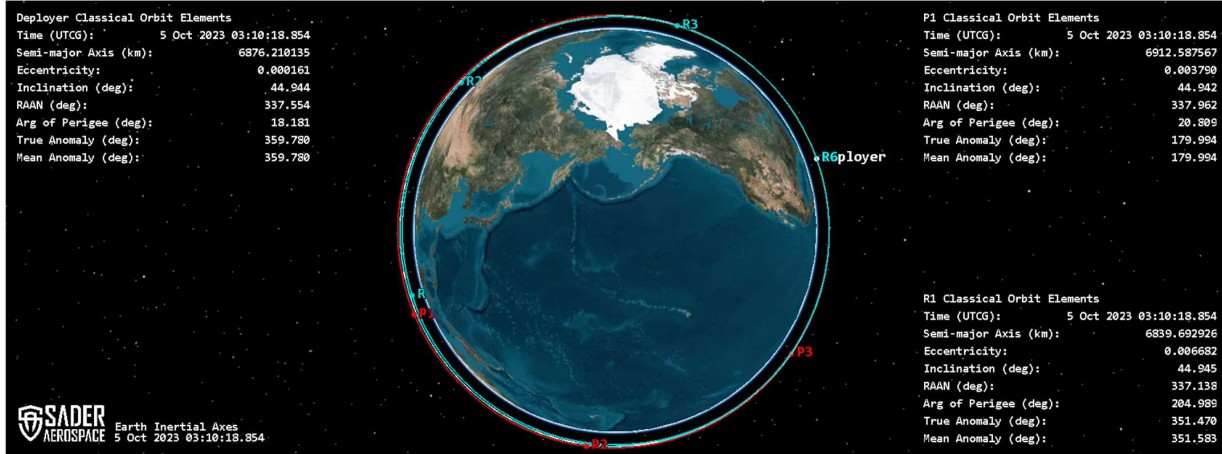

**Figure A4.** Constellation at the time of release maneuver 4 (5 October 2023 03:10:18.854). Objects P1–P3 (red) indicate the first through third prograde-released payloads, and objects R1–R3 (blue) indicate the first through third retrograde-released payloads.

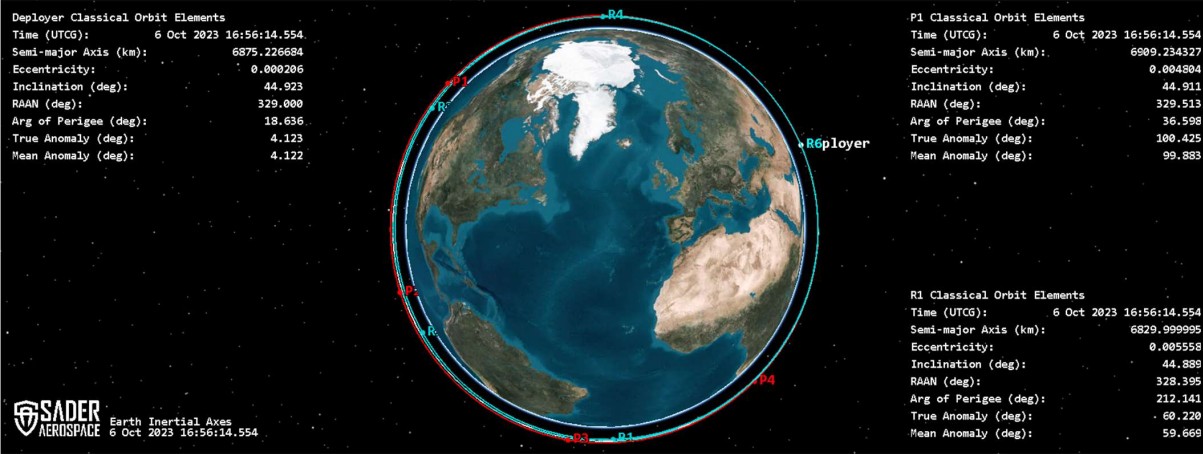

**Figure A5.** Constellation at the time of release maneuver 5 (6 October 2023 16:56:14.554). Objects P1–P4 (red) indicate the first through fourth prograde-released payloads, and objects R1–R4 (blue) indicate the first through fourth retrograde-released payloads.

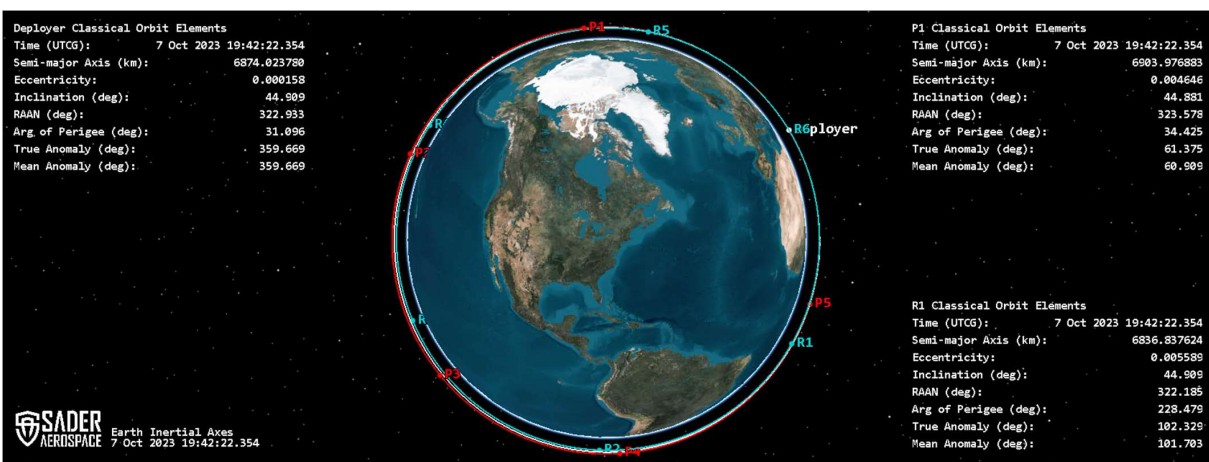

**Figure A6.** Constellation at the time of release maneuver 6 (7 October 2023 19:42:22.354). Objects P1–P5 (red) indicate the first through fifth prograde-released payloads, and objects R1–R5 (blue) indicate the first through fifth retrograde-released payloads.

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
