# Peer review of "Basic Orbit Design and Maneuvers for Satellite Constellations Deployed Using Momentum Exchange Tethers"

_aerospace, doi:10.3390/aerospace11030182_

Round 1
Reviewer 1 Report
Comments and Suggestions for Authors
Your study is interesting and will be useful in future space missions, but there are insufficient descriptions and unclear parts in the manuscript. Please check and revise according to the comments shown in the attached file.

Author Response
Thank you very much for your feedback. The following items have been added/modified in the newest draft, following the points you mentioned:
- Added a set of graphics that show MET systems in operation, see Figures 2-13 in the new draft.
- The graphic is just to show the MET concept on a CubeSat, tethers can either go with the payloads or stay with the deployer depending on how the designer wants to implement MET capabilities. While the concerns about drag and satellite lifetime are true, they aren’t in the scope of the manuscript, which is focused mainly on the initial deployment phase. Effects of the remaining tether system in later stages of a satellite’s life are going to be discussed in more detail in future work, but I have left some brief information in section 4.1.3 in the new draft that mentions that drag/lifetime effects should be kept in mind if a MET is to be used. For the section where you mentioned “Also, about the concept shown in Fig.1, may be, the precession occurs because the center of rotation is shifted from the center of mass, and it is difficult to insert satellites to the exact orbital planes. I understand the concept is just for the explanation of satellite release by SwingSat, but you need some explanations or proposal of concept that the readers do not have doubts.”, I am not quite sure what you are asking about, can you elaborate?
- For the usage of the software (STK), I have added some more details about the settings that were used in the new draft.
- The difference in altitude between ram and wake side payloads was included in the simulation. I have added information on this difference in the new draft. To help with clarification, I have also added a note saying that the maneuvers are general 20 m/s releases in each direction that assume ideal MET release vector alignment tangent to the deployer’s trajectory.
- We have separate calculation work for tether design which can be described, but to include it in this manuscript would require adding a new large section, (or an entire separate paper). This material is being planned to be included in a future publication. I have included some sample results from a custom tether design calculator that has been being developed. In the new draft, I have mentioned that this work is ongoing and will be described in future documentation.
- In short, for an example case for a single 500 kg payload being released at 20 m/s, some potential design options could be:
- Carbon fiber tape tether, 0.22 mm thick, 100 mm wide
- 10 m length, 2 rad/s rotation rate, .418 kg tether mass (plus ~0.252 kg steel spool shaft)
- 20 m length, 1 rad/s rotation rate, .836 kg tether mass (plus ~0.159 kg steel spool shaft)
- 40 m length, 0.5 rad/s rotation rate, 1.672 kg tether mass (plus ~0.100 kg steel spool shaft)
- Spring steel tape tether, 0.22 mm thick, 100 mm wide
- 10 m length, 2 rad/s rotation rate, 1.729 kg tether mass (plus ~0.252 kg steel spool shaft)
- 20 m length, 1 rad/s rotation rate, 3.458 kg tether mass (plus ~0.159 kg steel spool shaft)
- 40 m length, 0.5 rad/s rotation rate, 6.917 kg tether mass (plus ~0.100 kg steel spool shaft)
- Carbon fiber tape tether, 0.22 mm thick, 100 mm wide
- In short, for an example case for a single 500 kg payload being released at 20 m/s, some potential design options could be:
-
- For an example case for a single 50 kg payload being released at 20 m/s, some potential design options could be:
- Carbon fiber tape tether, 0.22 mm thick, 30 mm wide
- 10 m length, 2 rad/s rotation rate, .125 kg tether mass (plus ~0.00709 kg steel spool shaft)
- 20 m length, 1 rad/s rotation rate, .251 kg tether mass (plus ~0.00447 kg steel spool shaft)
- 40 m length, 0.5 rad/s rotation rate, .502 kg tether mass (plus ~0.00282 kg steel spool shaft)
- Spring steel tape tether, 0.22 mm thick, 30 mm wide
- 10 m length, 2 rad/s rotation rate, 0.519 kg tether mass (plus ~0.00711 kg steel spool shaft)
- 20 m length, 1 rad/s rotation rate, 1.038 kg tether mass (plus ~0.00450 kg steel spool shaft)
- 40 m length, 0.5 rad/s rotation rate, 2.075 kg tether mass (plus ~0.00285 kg steel spool shaft)
- Carbon fiber tape tether, 0.22 mm thick, 30 mm wide
- For an example case for a single 50 kg payload being released at 20 m/s, some potential design options could be:
- Added some color elements to improve the visibility on the figures in the new draft.
Reviewer 2 Report
Comments and Suggestions for Authors
The article provided a Satellite Constellations Deployed method using MET, which is interesting to readers.
I just suggest the author provide more detail literature review in introduction section, with more references in recent years.
Author Response
Thank you very much for your feedback. Material about some recent missions that used METs, along with some general history and methods of space tether usage has been added to the new draft.
Reviewer 3 Report
Comments and Suggestions for Authors
The main focus of this study is the development of a concept for deploying a satellite constellation using momentum exchange tethers. The author abstracts from tether deployment mechanisms, instead concentrating on describing constellation deployment options in terms of velocity impulse, orbital periods, and orbital elements based on mission requirements. The article contains a helpful diagram for determining which tether deploying method should be used in a given situation, making it of interest to readers. The article is written in clear and concise English. However, there are a number of comments on the work:
1. The figures are difficult to perceive due to the challenge of distinguishing between lines with different stroke frequencies. Perhaps using color in the diagrams could improve their clarity.
2. The technological readiness for designing and exploitation such systems is not clearly outlined in the Introduction and subsequent chapters. The choice of tether material and its impact on the dynamic aspects of constellation deployment should be addressed.
3. The article does not provide an explanation of how the velocity impulses can be calculated? what are the corresponding control laws? is arbitrary velocity impulses can be implemented by means of tethers? what is the time required for the control? The author should also include references to the relevant literature to address these issues.
4. There are insufficient references to the literature on the use of momentum exchange tethers for the deployment of satellite formations (not necessarily constellations). The author should cite and briefly comment on existing surveys of tether technologies and works on tether dynamics, explaining how they can contribute to the application of the proposed concept.
5. The term "symmetrical" is unclear in the context of the two sides of the tether from the deployment satellite. It may be beneficial to replace "symmetrical" with "synchronous" to avoid confusion.
It is recommended that the article be further revised to address these concerns.
Author Response
Thank you very much for your feedback. The following items have been added/modified in the newest draft, following the points you mentioned:
- Figures have been updated in the new draft to include colors.
- The new draft has some information added regarding the low technology readiness level of MET’s, and gives some initial thoughts on tether design/material. A statement is also added saying that the scope of the article is solely on the initial deployment process, and that tether design and later life effects are going to be described in future published work.
- Velocity impulses are produced by the release of a rotating payload/tether system, turning rotational kinetic energy into linear kinetic energy. The tangential velocity of the combined payload and tether’s center of mass is turned into the linear velocity change away from the deployer after release.
- In literature survey activities spanning approximately two years, there have been no results in finding existing material describing the combination of METs and constellations/satellite formations outside of works by the authors’ team. If you know of any potential resources though, we would be happy to look through them. The closest that has been found so far is material on deployments of individual payloads rather than multiple on a single flight. There is also some material on tethered formation-flying satellites (where the satellites are tethered together into certain formations for the duration of their lifetime), but that is a wholly separate concept from this as in that case, they are permanently connected and do not execute maneuvers away from one another in the same manner as the concept described in the manuscript, where they are rotated and released, permanently disconnected for the remainder of their lifetimes. Material about some recent missions that used METs, along with some general history and methods of space tether usage has been added to the new draft.
- In the manuscript, the original intention of the word “symmetric” relates to how in a symmetric deployment, the overall system’s geometry/configuration is symmetric (having a tethered payload on each opposing side of the deployer) as opposed to an asymmetric release (with only one tethered payload being deployed from one side of the deployer). I have added some more clarification on this situation in the new draft.
Round 2
Reviewer 1 Report
Comments and Suggestions for Authors
You almost responded appropriately to the comments, and the insufficient or unclear contents in the manuscript have been improved. So, it is considered that the manuscript has reached a level that can be published.